# LLM Agents Do Not Replicate Human Market Traders: Evidence from Experimental Finance

## Abstract

In this study, we compare Large Language Models (LLMs) with human traders in a classic experimental-finance paradigm where prices are determined endogenously. Using a well-established asset-trading design, we run homogeneous (mono-agent) markets with single-model LLM agents and heterogeneous "Battle Royale" markets with multiple LLM models. Our findings reveal that LLMs generally exhibit a "textbook-rational" approach, pricing the asset near its fundamental value and showing only a muted tendency toward bubble formation, whereas humans deviate substantially and consistently generate bubbles. Additional treatments, including dividend shocks and repeated-exposure ('experienced') runs, show that these differences persist across various experimental settings. Further analyses of LLM-generated strategy text indicate lower variance, reduced bias, and stronger reliance on fundamentals relative to humans' more heuristic-driven trading. These results highlight the risk of using LLM-agents to model human-driven market phenomena, as key behavioral features such as large, emergent bubbles are not reproduced.

## 1 Introduction

Financial markets are complex, dynamic coordination systems shaped by individual decisions made under uncertainty. While traditional financial theory assumes rational behavior, decades of behavioral finance research has revealed persistent biases, such as herding and overconfidence, that can give rise to mispricings and asset bubbles. These deviations underscore the challenges of achieving optimal outcomes in uncertain and evolving market environments.

The rise of agentic systems, particularly those driven by Large Language Models (LLMs), introduces a novel and previously unexplored dimension to this problem. As LLMs increasingly participate in domains traditionally navigated by humans, including financial systems and markets, understanding their behavioral tendencies has become critical. Questions such as; Do LLM-agents exhibit similar behavioral biases, behave more rationally, or fall victim to entirely different biases not seen in humans?

This paper presents the first comprehensive study comparing "out-of-the-box" LLM and human behavior within a single experimental finance paradigm. We adopt a classical trading environment where participants trade a risky asset with a controlled fundamental value and examine the behavior of LLM-agents in both homogeneous (mono-agent) markets, populated entirely by instances of the same model, and heterogeneous "Battle Royale" markets, where agents based on different base LLMs compete. These experimental markets are entirely endogenous, with prices determined solely by the buy-and-sell orders submitted by participants, and without any external price anchor or market maker. This enables us to dive deeper into how the underlying agent behavior affects the market, as we can control all market parameters.

Previous research offers conflicting perspectives on AI-human behavioral alignment. Some works suggest a significant difference between human and LLM interactions (Anlló et al. (2024)), while others indicate that LLM behavior could supplement or even substitute for human studies (Leng (2024); Horton (2023); Jia et al. (2024)). We address this tension in the literature through rigorous experimental comparison in experimental finance, a domain that has long served as a testbed for understanding financial decision-making under risk and uncertainty.

**Contributions** Our findings reveal clear behavioral differences between LLM and human-driven markets. LLM markets demonstrate substantially more "textbook-rational" behavior than their human counterparts. Human markets consistently generate significant price bubbles and deviations from the asset's fundamental value, while LLM-agents generally trade much closer to fundamental value and exhibit markedly reduced trading variance. We also find that LLMs appear to incorporate less bias into their future price forecasts and employ strategies more focused on fundamental value, rather than strategies commonly used by humans. These results challenge the notion that "out-of-the-box" LLMs can faithfully replicate human market dynamics, particularly in terms of the emergence of market-level phenomena such as bubbles and crashes. As agentic systems increasingly participate in and influence real financial markets, understanding these behavioral divergences becomes essential for effective regulation, risk management, and market design.

## 2 BACKGROUND

Numerous works have studied LLMs in strategic environments such as social dilemmas and iterated games Brookins & DeBacker (2024); Horton (2023); Akata et al. (2023). In finance, most existing studies focus on empirical applications—LLMs trading in real markets or analyzing financial data (e.g., Xiao et al. (2024); Li et al. (2024); Yu et al. (2024a); Yun (2024); Liu & Lo (2025); Ding et al. (2024); Abdelsamie & Wang (2024); Chen (2025); Zhou et al. (2025)). However, to our knowledge, no prior work has analyzed LLM behavior in experimental markets where prices are endogenously generated by agent interaction (see Appendix C for a more comprehensive review).

In finance, LLMs have regularly been applied to real-world scenarios. Papers such as; Zhang et al. (2023); Bond (2023); Kirtac & Germano (2024) explored LLMs' ability to process financial sentiment and predict market movements. Fatouros et al. (2024) introduced MarketSenseAI, which leverages GPT-4 for stock selection, illustrating the utility of LLMs in navigating complex financial data. While these studies demonstrate the promise of LLMs in financial settings, there is still a need to understand how these agents perform in experimental financial settings where strategic interactions drive outcomes.

The study of market dynamics in controlled experimental environments has revealed critical insights into behavioral biases and inefficiencies. For instance, Smith et al. (1988) demonstrated the formation of bubbles and crashes in experimental markets, even under conditions of perfect information. Subsequent work by Bostian & Holt (2009) highlighted how individual behaviors, such as overconfidence and herding, contribute to these phenomena. Incorporating AI agents into such paradigms presents a unique opportunity to evaluate whether these systems replicate, mitigate, or exacerbate market inefficiencies.

## 3 EXPERIMENT

We focus on a classic experimental finance paradigm with a constant fundamental value, the mathematical price of an asset; see Equation B in the Appendix. We adopt the experimental design from Smith et al. (2014) and Holt et al. (2017), which involves an endogenous experimental asset market where subjects trade a risky asset (STOCK) in an open-call market for 30 periods operated using the oTree experiment platform (Chen et al. (2016)). Participants are given 4 shares of STOCK and 100 units of CASH. Participants have two options: either invest CASH in STOCK, which in each trading period pays out dividends of either 0.4 or 1 units of CASH with equal probability, or hold CASH, which generates interest at 5% each trading period. The price is determined solely by the trading activity of the experimental market participants.

After the experiment, the risky asset is redeemed for a fixed value of 14 units of CASH, representing the fundamental value of the STOCK (See Appendix B for full formulation).

Because of these parameter decisions (i.e., dividend schedule, CASH interest rate, and redemption price), the fundamental value remains constant throughout the entirety of the experiment, enabling us to objectively classify when the market is in a "bubble" state Bostian et al. (2005). This differs starkly from applied financial studies, where the fundamental value of a real-world asset is unknown. Further discussion of the experimental details can be found in Appendices A.

This task was assigned to both LLM agents and human traders in separate markets. In adapting the task to LLM-agents, our primary goal was to best emulate the experiment presented to human subjects. The description of the task setting mirrors the original instruction video, shown to human participants (See Appendix H.2). In every round of the experiment, each agent simultaneously submits orders to the market as well as expectations regarding current and future market prices.

A common challenge in adapting multi-round games to LLM-agents is the persistence of memory between rounds, which is critical to learning and adapting strategies. Building off previous works (Fish et al. (2024)), we allow agents to communicate with future versions of themselves by allowing them to read and write their own "Insights" and "Thoughts" at each round. This structure incorporates Chain-of-Thought (CoT) reasoning (Wei et al. (2022)), which has been shown to help elicit the reasoning ability of LLMs, providing an interpretable window into how agents perceive and interact with the market. Further prompting details are presented in Appendix I.

To frame our analysis, we categorized potential outcomes into three groups: **rational (R)**, where models do not generate bubbles and trade close to the fundamental value; **human (H)**, where models generate bubbles in a manner similar to humans; and **erratic (E)**, where models perform inconsistently, failing to adhere to any consistent strategy, whether rational or human-like.

## 4 SINGLE-MODEL LLM MARKET PERFORMANCE

In this section, we evaluate how single-model LLM markets behave along two critical dimensions: rational adherence to fundamental asset value and similarity to human market dynamics. Table 1 and Figure 1 summarize our key findings.

### 4.1 EXPERIMENTAL SETUP

We conducted three markets using agents based on six different API-based commercial LLMs: Anthropic's Claude-3.5-Sonnet Anthropic (2024), OpenAI's GPT-3.5 and GPT-4o OpenAI (2025), xAI's Grok-2 xAI (2024), Mistral-Large AI's Mistral-Large AI (2024), and Google's Gemini-1.5 Pro DeepMind (2024). Each market consisted of 20 identical instances of the same LLM agent. The agents were given the same prompt and were unaware of the identities of their counterparts. Each instance was instructed to maximize its profit.

For comparison, we also ran nineteen human-only markets, with identical experimental conditions (though the humans also conducted a short risk-elicitation task in between each round). Participants were a mix of [UNIVERSITY] students and staff, as well as online participants recruited through `prolific.com`. The human participants were given instructions that mirror the LLM prompts; full instructions are shown in Appendix section H.2. The average plot from these experiments is shown alongside the LLM-only Battle Royale market results in Figure 2.

#### 4.1.1 SINGLE-MODEL MARKET ANALYSIS

As shown in Table 1, Claude-3.5 Sonnet and GPT-4o yield the lowest MSEs and nearly zero correlation with human price paths, keeping prices close to fundamentals and mapping to the Rational (R) hypothesis. Both also show low portfolio variance, suggesting consistent strategies across instances.

By contrast, Grok-2 and GPT-3.5 show the largest deviations from fundamentals and the highest correlations with human markets. Grok-2 exhibits bubble–crash cycles, while GPT-3.5 generates a monotonic run-up without correction (Figure 1), a pattern occasionally seen in humans.

Mistral-Large also produces bubbles but with smaller peaks and sharper crashes, dipping below fundamentals late in the session. While more rational in absolute terms (MSE $\approx$ 5.7 vs. 429.8 for humans), its dynamics align with the Human (H) hypothesis.

Finally, Gemini-1.5 Pro departs from both rational and human-like behavior, trading well below fundamentals before correcting upward. This so-called "reverse bubble" (negative PCC) suggests erratic dynamics rather than coherent strategy, mapping most closely to the Erratic (E) hypothesis.

Table 1: Market performance metrics

| Model | MSE (Fundamental) | PCC (Human Avg) | Closest Hypothesis | PV Variance |
|---|---|---|---|---|
| Claude-3.5-Sonnet | 0.536 | 0.001 | R | 26.15 |
| GPT-4o | 0.789 | 0.050 | R | 22.76 |
| Grok-2 | 17.325 | 0.558 | H | 39.43 |
| Mistral-Large | 5.694 | -0.112 | H | 49.96 |
| GPT-3.5 | 26.367 | 0.490 | H | 30.44 |
| Gemini-1.5 Pro | 3.103 | -0.401 | E | 45.58 |

**Notes: MSE vs. Fundamental Value** measures rationality by assessing how closely the market prices align with the fundamental value. **PCC vs. Human Market Average** quantifies the similarity of LLM-driven markets to typical human market price trajectories using the Pearson correlation coefficient (PCC). **Portfolio Value (PV) Variance** reflects strategy diversity among agents, indicating whether they adopt varied approaches or behave uniformly. The **Closest Hypothesis** column maps each model's behavior to the closest hypothesized market behavior pattern, based on a mix of quantitative and qualitative comparisons. R if MSE around fundamental was below 1, and H if we observed a bubble formation and crash, E if the market behavior was nonsensical (reverse bubble).

Figure 1: Mono-Agent Price Charts compared to Humans

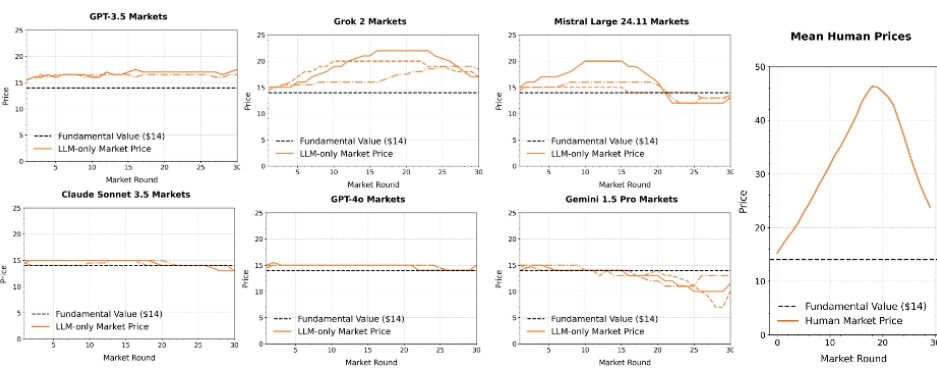

None of the LLM-agents generate market bubbles similar in magnitude to humans.

# 5 BATTLE ROYALE: COMPARING LLM MARKET PERFORMANCE

To evaluate how various LLM agents fare when competing in the same market, we conducted *Battle Royale* market experiment with a heterogeneous mixture of the models. By introducing diverse LLMs into a single environment, we aimed to explore:

- Which LLM is most effective at maximizing profits in a competitive, multi-agent setting? Which Agent is the best trader?
- How a mixed-LLM market behaves, and whether it will exhibit emergent phenomena when models with distinct trading strategies interact?
- Whether heterogeneous LLM interpretation and strategy differences are enough to generate financial bubbles?

## 5.1 EXPERIMENTAL SETUP

We conducted markets involving the same six LLM models as in the Single-Model section. Each market included a total of 24 agents (four agents per model). The agents operated under identical conditions and were tasked with maximizing their profits over 30 trading periods. The risky asset in these markets maintained a constant fundamental value of 14 units, consistent with prior experiments. Figure 2 shows the three 'Battle Royale' market results.

## 5.2 BATTLE ROYALE MARKET ANALYSIS

Across three heterogeneous markets, two converged on prices near fundamentals, while one produced a modest bubble resembling human markets but with a lower peak. Trading volume spiked at

Figure 2: Battle Royale Market Performance with Human Comparison

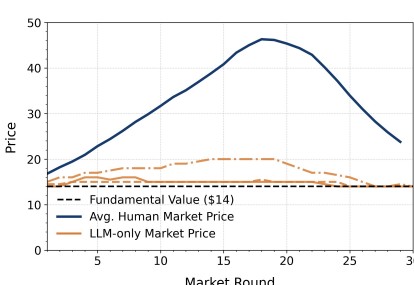

Table 2: Battle Royale Agent Performance

| Model | Avg Portfolio Value $\pm$ 1 STD |
|---|---|
| Mistral-Large | $689.56 \pm 4.11$ |
| Gemini-1.5 Pro | $688.49 \pm 4.48$ |
| Claude-3.5 Sonnet | $680.36 \pm 5.08$ |
| GPT-3.5 | $674.42 \pm 3.48$ |
| GPT-4o | $671.36 \pm 17.30$ |
| Grok-2 | $668.51 \pm 27.01$ |

| **Market-Level Summary Statistics** | |
|---|---|
| MSE (Fundamental): **5.86** | PCC (Human Avg): **0.27** |

the end of this bubble run, echoing the late-round rush seen in human experiments (Figure 2). Overall, these dynamics align most closely with the Rational (R) hypothesis, although they demonstrate that competitive interaction can occasionally generate bubble-like patterns.

At the agent level (Table 2), portfolio values cluster tightly, with no model consistently dominating. The spread between best and worst traders remains small, suggesting convergence on broadly similar strategies and limited inequality of outcomes in mixed markets. This shows that none of these models are able to consistently best the others in competitive trading.

# 6 TRADING STRATEGY ANALYSIS

After each round, LLM agents were prompted to provide details of the strategy they used to drive their trading behavior. This was done both to allow us to analyze their behavior more deeply and to enable the LLMs to develop long-running strategies, utilizing Chain-of-Thought-like reasoning. In contrast, human subjects were asked what approach they employed only at the end of the experiment. More details of our data cleaning techniques and additional analysis can be found in Appendix F.

We first examined the word frequency of both humans and LLMs in the mono-agent runs. Of the top six words for each group, five are the same, though in a different order ("sell", "buy", "stock", "price", and "market"; see Table 6 in Appendix for details). The top ten most popular words for the human subjects include "low" and "high", which do not appear among the top ten words used by the LLMs. Conversely, the words "continue" and "buyback" are among the most common for the LLMs (see also word clouds in Figure 22 in the Appendix). This pattern suggests that human traders follow a "buy low, sell high" strategy in an attempt to outperform the market. In contrast, LLM agents adopt a longer-term approach that emphasizes the asset's intrinsic value.

## 6.1 TEXT MINING AND STRATEGY COMPARISON TO HUMANS

To assess whether LLMs engaged in coordination or converged on shared narratives, we ran a text–mining analysis of the "insight" and "plan" files. We found no explicit coordination language (0 hits on terms such as "coordinate" or "work together"), and only a handful of generic references to "anticipating other agents." By contrast, we observed several instances of near–orthogonal language between agent pairs (cosine similarity $\leq$ 0.1; e.g., "buy aggressively" vs. "sell off"), suggesting divergence rather than cooperation. Overall, mixed–model markets appear to consist of independent traders reacting to shared price signals rather than colluding.

We then directly compared LLM and human strategies. A classifier trained on the insights/plans text achieved >99% accuracy in identifying which model produced which text. Extending this to a seven–way classifier including humans (using their end of experiment reflections), we found that ~33% of Gemini-1,5 Pro and GPT-4 reflections were misclassified as human, while all other models were below 10%. Binary classifiers confirmed this pattern: GPT-4 text was misclassified as human nearly 50% of the time (Mistral-Large 36%; GPT-3.5 28%), while other models were below 3%. Despite superficial stylistic overlap, cluster analysis of strategy words showed almost no clusters containing both humans and LLMs (in one run, only 8/497 humans clustered with any model), suggesting that the underlying reasoning remains distinct.

To probe this distinction further, we ran a Latent Dirichlet Allocation (LDA) Blei et al. (2003) on all strategies combined. Using the Python `gensim` library Řehůřek & Sojka (2010), we assigned each strategy to one of two topics. The first topic, identified by the words *orders, sell, buy, shares, stock*, denoted strategies aligned with a "buy low; sell high" practice. The second topic, identified by the words *stock, adjust, buyback, closely, current*, reflected a more fundamental value focused approach. The LDA assigned 89.3% of human strategies to the first topic but only 36.6% of the LLM strategies; conversely, 63.4% of LLM strategies (10.7% of humans) fell into the second category. A t-test confirmed this difference is highly significant ($p < .001$). Figure 23 shows this distribution.

## 6.2 LINGUISTIC AUDIT OF MODEL STRATEGIES

We also wanted to identify why specific models generate larger, more human-like bubbles, while others adhere relatively closely to fundamentals. To do so, we ran a finer-grained linguistic audit of the "insight" and "plan" outputs. Each statement was tagged as *speculative*, *fundamental*, or *generic* based on keyword dictionaries.

Table 3: Distribution (%) of speculative, fundamental, and generic text in agent strategy statements. The behavior column reflects observed market outcomes.

| Model | Speculative | Fundamental | Generic | Behavior |
|-------|-------------|-------------|---------|----------|
| Grok-2 | 68.4 | 16.0 | 15.6 | Bubble-prone |
| GPT-3.5 | 28.4 | 18.0 | 53.6 | Bubble-prone (No Crash) |
| GPT-4o | 17.9 | 59.6 | 22.5 | Bubble-dampening |
| Claude-3.5 Sonnet | 14.5 | 82.8 | 2.7 | Bubble-dampening |
| Gemini-1.5 Pro | 20.5 | 58.7 | 20.8 | Bubble-dampening |
| Mistral-Large | 35.8 | 52.0 | 12.2 | Mixed |

**Speculative keywords:** "trend", "momentum", "price rising", "price falling", "buy more", "bubble", "sell off", "rally".
**Fundamental keywords:** "undervalued", "intrinsic", "dividend", "fair value", "discount", "fundamental", "14", "buy back".

Grok-2 and GPT-3.5 seem to create bubbles because speculative narratives dominate their reasoning, whereas GPT-4o, Claude-3.5, and Gemini-1.5 Pro seem to dampen exuberance by emphasizing intrinsic value anchors. Mistral-Large oscillates between the two, producing smaller, transient bubbles with a trajectory of value → momentum → uncertainty. These linguistic signatures provide a possible mechanistic explanation for the divergent market outcomes observed across models.

## 7 MARKET TREATMENTS

Beyond the baseline constant–fundamental design, we tested whether LLM's rationality profiles held under more complex and dynamic market conditions, such as when payoffs and fundamental values shift over time.

To test whether our findings extend beyond constant fundamental value, we conducted two separate mono-agent market experiments (each with three sessions) featuring a mid-experiment dividend shock. Beginning in Round 15, we either doubled (7.1) or halved (7.2) the dividend and redemption value, which doubles/halves the fundamental value of the asset. The agents were not informed of this change prior to the experiment and were told at Round 15 that the dividend and redemption value had changed (either doubled or halved) and would remain so for the rest of the experiment. This mirrors classic shock tests in experimental finance, and simulates how unexpected "news" can impact the value of an asset, allowing us to assess whether LLMs adjust appropriately when the fundamental value changes. Note we were unable to run Grok 2 as the public endpoint has been deprecated.

### 7.1 DIVIDEND ("NEWS") SHOCK DOUBLING

Results show that all models except GPT-3.5 quickly converged toward the new fundamental value. Claude-3.5 Sonnet, GPT-4o, Gemini-1.5 Pro, and Mistral-Large all incorporated the new information, though the process was not instantaneous, stabilizing prices near $28. GPT-3.5, however, lagged behind: its forecasts and orders remained anchored near the pre-shock equilibrium. Notably, Gemini-1.5 Pro occasionally overshoots the FV, creating minor bubbles and crashes.

Figure 3: Mono-Agent Market Price Charts Dividend Shock Doubling

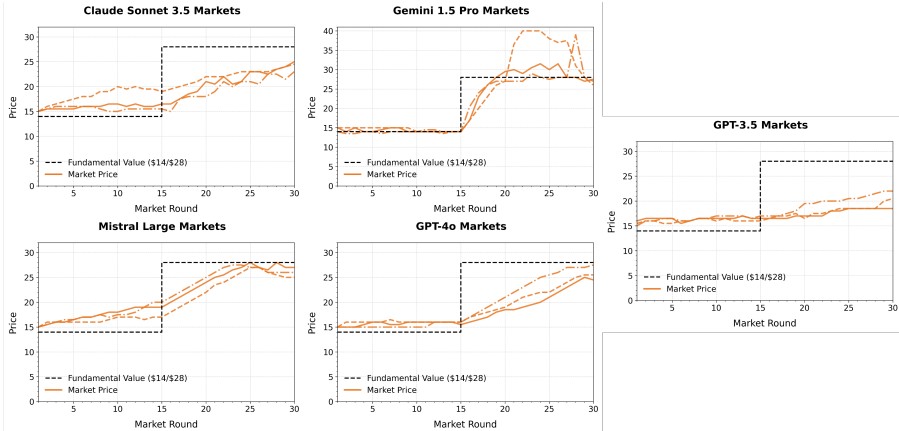

Models slowly adjust to the new FV price, with the exception of GPT-3.5. We observe that Gemini-1.5 Pro occasionally creates a modest bubble.

## 7.2 DIVIDEND ("NEWS") SHOCK HALVING

As shown in Figure 4, all models except GPT-3.5 converged toward the new fundamental value. Claude-3.5 Sonnet, Gemini-1.5 Pro, and Mistral-Large all incorporated the new information within 2–3 rounds, stabilizing prices near $7. GPT-3.5 and GPT-4o, however, lagged behind, with GPT-3.5's prices remaining anchored near the pre-shock equilibrium, and GPT-4o incorporating the information slowly.

Figure 4: Mono-Agent Market Price Charts Dividend Shock Halving

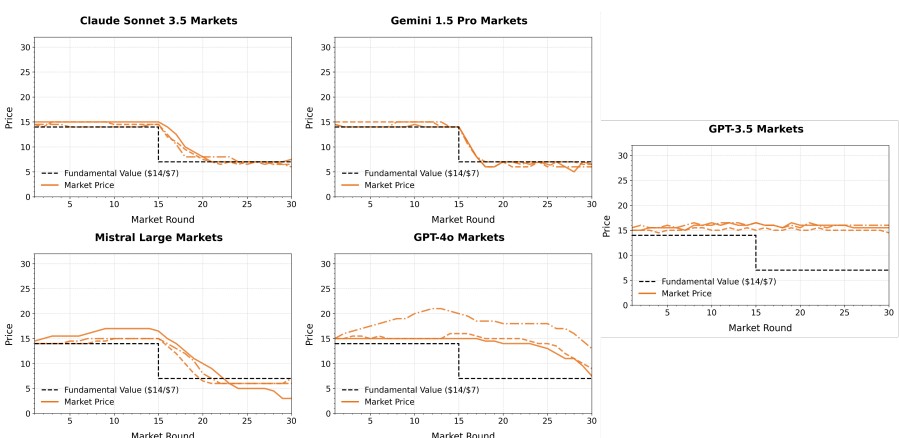

All of the markets, with the exception of GPT-3.5, quickly adjust to the new fundamental value after the dividend shock. We observe that GPT4 operates within a modest bubble before converging to its redemption value towards the end of the market session.

These treatments reinforce our main result: most LLMs exhibit a fundamental value oriented adjustment, while GPT-3.5 is uniquely prone to inertia and bubble-like dynamics. We notice a slight increase in the propensity for the models to create bubbles, but they still occur infrequently. It may appear that the models converge more quickly to downward shocks than upward shocks. Please keep in mind that upward shocks are larger in absolute magnitude, and our experiment design limits orders to ± $3 of the current price. This was done to make the experiment accessible to humans and carried over to our LLM experiments for parity.

We also conduct multiple robustness checks in the Appendix, ensuring that slight experimental differences (such as removing the risk-elicitation task for LLMs D.2) or LLM trading experience

(D.1) do not materially alter our conclusion. While these treatments represent only a small subset of the numerous possible experimental modifications, they demonstrate that our conclusion regarding LLM rationality remains robust to various changes in market conditions.

# 8 RATIONALITY PROPERTIES OF INDIVIDUAL LLM FORECASTS

In addition to analyzing the pricing behavior of LLMs, we were also interested in how these models generate forecasts for future price periods. We captured each model's forecasts, which allowed us to assess not only the rationality of their forecasts, whether they align with fundamental valuation principles, but also whether the LLMs themselves behave rationally in response to their expectations. We compare this analysis to that of our human traders to understand whether LLMs, in the mono-agent market setting, make similar or different forecasting mistakes compared to their human counterparts. The mathematical formulation for calculating the mean and median forecasts for any given session is provided in the Appendix (Section G.3).

Forecast errors are defined as $E_{i,t}^h = P_{t+h} - f_{i,t}^{t+h}$, where $P_{t+h}$ denotes the actual price at round $t + h$ and $f_{i,t}^{t+h}$ is agent $i$'s forecast formed at round $t$ for horizon $h$; positive values indicate underprediction. Full distributions and horizon-specific test results are reported in Appendix G.3. Humans exhibit substantial systematic underprediction: the mean one-step error ($h = 1$) is $1.67$. By contrast, all LLMs' mean errors are close to zero. Consistent with this, Appendix Figure 26 shows that LLM forecast-error distributions are more tightly centered around zero than those from human markets, indicating higher forecast precision.

Figure 5: Mean Forecast Error by Round

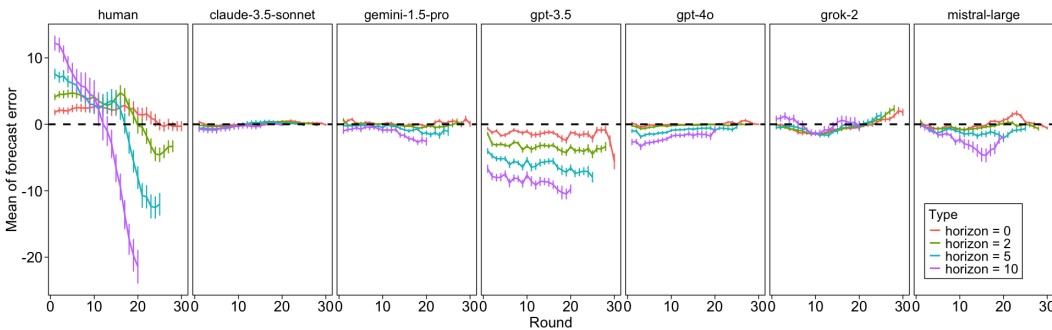

Humans exhibit substantial systematic underprediction, with a mean one–step forecast error of $1.67$. By contrast, all LLMs are much closer to zero.

We also applied three additional diagnostics to compare the forecast-rationality profiles of LLM agents and human subjects: (i) **unbiasedness**—whether forecast errors are centered on zero (no systematic bias); (ii) **zero autocorrelation**—whether errors in one round fail to predict errors in later rounds; and (iii) **errors uncorrelated with forecasts**—whether the forecast itself fails to predict the size of the error (no optimism/pessimism bias).

Results are summarized in Table 4 and expanded upon in Section G.3 of the Appendix. Results suggest that LLMs generally forecast more accurately than humans in the short run, but their rationality profiles differ between models. GPT-3.5 is highly biased and correlated; Claude-3.5 and GPT-4o are the most consistently rational; Gemini-1.5 Pro, Grok-2, and Mistral-Large fall in between. Importantly, no model fully reproduces the human error pattern.

# 9 POTENTIAL LIMITATIONS AND BROADER IMPACTS

Our results should be interpreted in light of several caveats. First, LLM agents do not experience incentives, emotions, or online learning in the human sense; their "strategies" emerge from the prompt structure and frozen weights, rather than endogenous reasoning or reinforcement. Second, we study stylized markets with a single asset and a short horizon. How LLM traders respond to

Table 4: Forecast Accuracy and Rationality Properties of Humans and LLMs.

| Model | Forecast Error | Unbiasedness | Zero Autocorr. | Errors $\perp$ Forecasts |
|---|---|---|---|---|
| Humans | 1.67 | 0.56 | 0.17 | 0.31 |
| Claude-3.5 Sonnet | 0.09 | 0.46 | 0.08 | 0.16 |
| Gemini-1.5 Pro | 0.14 | 0.41 | 0.18 | 0.41 |
| GPT-3.5 | −1.54 | 0.00 | 0.60 | 0.00 |
| GPT-4o | 0.04 | 0.40 | 0.60 | 0.03 |
| Grok-2 | −0.31 | 0.56 | 0.18 | 0.27 |
| Mistral-Large | 0.00 | 0.44 | 0.14 | 0.36 |

**Notes:** "Forecast Error" is the mean one–step error ($E_{i,t}^0$; positive = underprediction). Other columns report the proportion of agents passing each rationality test, averaged across horizons $h \in \{0, 2, 5, 10\}$. Full horizon-specific results are reported in the Appendix.

longer horizons, short selling, or richer strategic interactions remains an open question. Finally, we evaluated only a handful of foundation models without fine-tuning. Model choice, prompting tactics, or retrieval-augmented memory may materially alter behavior.

Despite these limitations, our work has three areas of immediate impact. First, in live markets, several hedge funds have begun deploying LLM-driven order flow. As machine agents come to represent a measurable share of trading volume, both regulators and practitioners will need empirical baselines to understand the dynamics they generate. Second, almost all significant findings in behavioral finance (loss aversion, herding, mental accounting) were initially discovered through lab experiments (Kahneman & Tversky (2013), Thaler (1985)). As we attempt to understand how LLMs trade and the biases they exhibit, we thought it best to adopt the same approach. Third, for experimental design, recent papers have explored the idea of substituting LLMs for human subjects in laboratory finance experiments (Rio-Chanona et al., Cui et al.); our evidence of systematic departures from human behavior underscores the risk of doing so without rigorous validation. Therefore, we advise caution in replacing human subjects with "off-the-shelf" LLM agents. We acknowledge these limitations and thus view this study as a *starting point, rather than an endpoint, for incorporating LLM agents into financial and behavioral research*.

## 10 DISCUSSION AND CONCLUSIONS

In this study, we compare the behavior of Human and Large Language Models (LLMs) in an experimental finance paradigm that has historically elicited asset bubbles and crashes when conducted with human participants. We found that LLM-agents typically converge on prices near the fundamental value, thereby exhibiting more "textbook-rational" behavior than human subjects. Although some single-model markets did display mild bubble-like phenomena, particularly GPT-3.5, which inflated prices without a clear crash, and Mistral-Large and Grok 2, which produced modest run-ups, these tendencies were weaker than what is commonly observed in human-only markets. Moreover, individual outcomes in LLM-driven markets tended to cluster more tightly around similar final portfolio values, indicating less strategic heterogeneity than seen among humans. We found that these findings persisted even under changing market conditions, news/dividend shocks. These findings collectively suggest that although LLMs may serve as rational traders, they seldom replicate the pronounced boom-and-bust patterns that emerge from human behavioral biases. We also analyze the forecasting behavior of these LLM agents and find that the majority of the LLMs generate more accurate forecasts of future price behavior than human traders; however, we observe some heterogeneity among LLM agent models in their ability to forecast rationally. Across the board, the LLM agents performed differently from humans on most tasks; therefore, our results caution against using LLMs as replacements or for piloting for human participants in experimental finance studies. Future work may explore whether new prompting strategies or incentives can induce more human-like trading patterns from LLM-agents. However, for now, it remains clear that these models trade with a consistency and rationality that diverge significantly from the behaviors displayed by human counterparts.

## 11 ETHICS STATEMENT

We do not believe there are any ethical concerns with this work, all human data was collected under review of the IRB at [University]. The experiments with large language models involved only synthetic agents and did not expose participants or sensitive populations to potential harm. All analyses follow standard practices in experimental economics and machine learning. We believe this work adheres fully to the ICLR Code of Ethics and raises no issues regarding privacy, fairness, security, or potential misuse.

## 12 REPRODUCIBILITY STATEMENT

All code will be released alongside the camera-ready submission. Complete experimental details, including instructions provided to agents and human participants, as well as the full set of prompts, are documented in Appendix A and Appendix I. These materials, together with the statistical procedures and robustness checks reported in Section 4 and Appendices E–G, ensure that all results can be readily reproduced. Our goal is not only to enable replication of the experiments presented here, but also to provide a foundation that future work can extend and build upon.

## 13 LLM DISCLOSURE

Large Language Models (LLMs) were used in a limited capacity to assist with LaTeX formatting and debugging. They were not used for research ideation, experimental design, data analysis, or substantive writing. The authors take full responsibility for the content of this paper.

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

# A   ADDITIONAL EXPERIMENT DETAILS

We now elaborate on the details of the experimental setup. There are 3 practice trading rounds followed by 30 rounds of trading in the main experiment. The practice rounds have no impact on the main experiment rounds and serve only to familiarize participants with the market setting. During each round, participants interact with the market through a sequence of screens.

Our experimental design involves an asset market where subjects trade a risky asset in an open-call market for 30 periods (Smith et al., 2014; Holt et al., 2017). Participants are given 4 shares of stock and 140 units of cash. Participants have two options: either invest cash in the risky asset, which pays out dividends of either 0.4 or 1 with equal probability, or hold cash, which accrues interest at a rate of 5% each trading period. The dividend payment is drawn independently for each round.

The experiment consists of a sequence of four screens in this order, as depicted in the Figures below:

1. Order Submission Screen (20 seconds)(Figure 6): Participants submit orders to buy or sell the asset

Figure 6: Order Submission Page

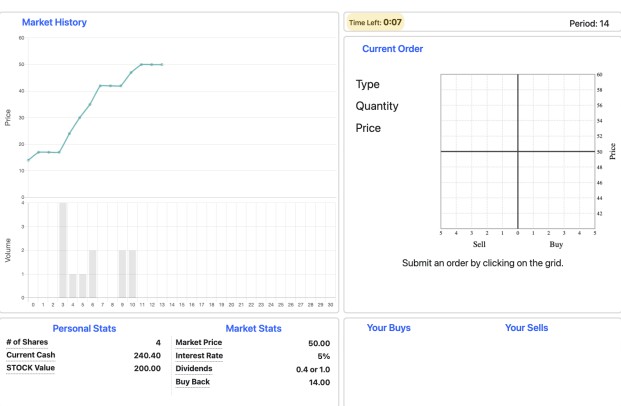

2. Forecast Screen (30 seconds)(Figure 7): Participants submit their expectations regarding current and future market prices.

Figure 7: Forecast Page

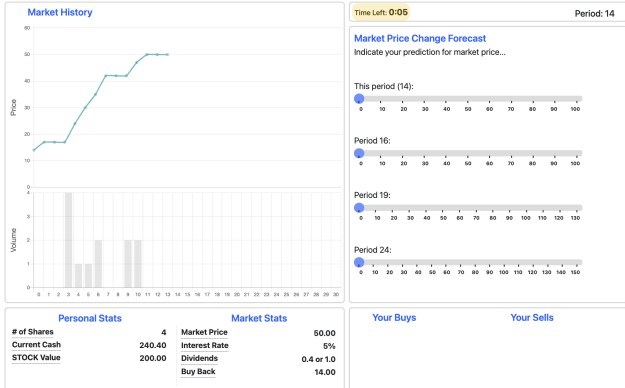

Participants are asked to submit their estimation for the current market period and periods 2, 5, and 10 rounds ahead.

3. Round Results Screen (10 seconds)(Figure 8): Participants see the actual realized price for that period, how many shares were traded in total, and how many shares they traded.

Figure 8: Round Results Page

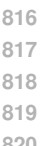

After the experiment, the risky asset is redeemed for a fixed value of 14 units, representing the fundamental price, calculated using the formula presented in Equation 7. Note that our experiment paradigm was designed such that this fundamental value remains constant throughout the entirety of the experiment. This enables us to objectively classify when the market is in a "bubble" state, given we always have a constant fundamental value to compare the market price to.

## A.1 PRICE FORMATION

We now describe the procedure used in computing the market-clearing price for the call auctions in our experiment. For every session $s$ and round $r$, the market price is determined as the price that maximizes volume Sun et al. (2010). A bid submitted by subject $i$ in round $r$ of session $s$ is a tuple consisting of price and quantity, $(p, q)$. Here, $i = \{1, 2, 3\}$ indicates that a subject may submit up to 3 bids per round. The set of all bids submitted in a given market session and round is represented by $B_{s,r}$. Likewise, the set of all asks is denoted by $A_{s,r}$, with similar notation for individual asks, $(p, q)$.

Let $P_{s,r}$ be the set of all prices submitted as bids or asks,

$$P_{s,r} = \{p \mid (p, q) \in B_{s,r} \cup A_{s,r}\}, \tag{1}$$

where $p((p, q))$ is a function that returns the price portion of a given order tuple (bid or ask).

Define the cumulative buy quantity of a given price $p$ as the total quantity of all bids at or above that price:

$$Q^B(p) = \sum_{(p', q) \in B_{s,r}} q \cdot \mathbb{1}_{p' \geq p}, \tag{2}$$

where $q((p, q))$ is a function that returns the quantity portion of a given order tuple (bid or ask). This represents the total quantity of shares that all participants are willing to purchase at a given price.

The cumulative sell quantity is likewise defined as:

$$Q^A(p) = \sum_{(p', q) \in A_{s,r}} q \cdot \mathbb{1}_{p' \leq p}, \tag{3}$$

and is the total quantity of shares that all participants are willing to sell at a given price.

For a given price $p$, volume is determined by

$$V(p) = \min(Q^B(p), Q^A(p)). \tag{4}$$

The market price is then

$$p^*_{s,r} = \arg \max_{p \in P_{s,r}} V(p). \tag{5}$$

All prices are rounded down to the nearest integer value, a consideration made to simplify the user interface.

If there is no intersection between the bid and ask, the mid-point between the highest bid and the lowest ask is reported as the market price for that period.

## A.2 COMPUTE DETAILS FOR REPRODUCIBILITY

For our experiments, we leverage commercially available API endpoints from Anthropic's Claude-3.5-Sonnet (Anthropic, 2024), Google's Gemini 1.5 Pro (DeepMind, 2024), Mistral's Mistral-Large 24.11 (AI, 2024), OpenAI's GPT-3.5 and GPT-4o (OpenAI, 2025), and xAI's Grok-2 (xAI, 2024) to underpin the agents trading in our markets.

We note that our results are reproducible only insofar as these models remain accessible via their respective APIs. To contextualize the compute requirements for our experiments, Table 5 reports the approximate token usage per model. These figures include tokens consumed during pilot runs, which primarily involved the OpenAI and xAI models. Token usage for Gemini 1.5 Pro could not be accessed through the API console.

| Provider | Model | Approx. Token Usage |
|---|---|---|
| Anthropic | Claude-3.5-Sonnet | 17.9M |
| Google DeepMind | Gemini 1.5 Pro | Not Available |
| Mistral | Mistral-Large 24.11 | 18.7M |
| OpenAI | GPT-3.5-turbo-0125 | 30.1M |
| OpenAI | GPT-4o-2024-08-06 | 24.7M |
| xAI | Grok-2 | 33.9M |

Table 5: Approximate token usage across models.

## B FUNDAMENTAL VALUE CALCULATION

For completeness, we present a derivation of the fundamental value (FV) of the asset trading in our experiments. We show that (1) the FV is constant and (2) the FV of the asset matches exactly the redemption value. We assume minimal familiarity with financial economics terminology.

In each period, a unit of stock pays a stochastic dividend drawn i.i.d. from a Bernoulli distribution over $\{0.4, 1.0\}$, with equal probability. Therefore, the expected dividend is:

$$\mathbb{E}[D_t] = 0.5 \cdot 0.4 + 0.5 \cdot 1.0 = 0.7$$

Cash holdings earn a constant risk-free return of $r = 5\%$ per period. Consequently, the present value of receiving a dividend payment $D_t$ (or any transfer in general) some $t$ rounds in the future is:

$$\frac{D_t}{(1+r)^t}$$

Let $T$ be the final trading period, and suppose the asset is redeemed at time $T$ for a fixed amount $V_T = 14$. Then the fundamental value of the asset in any round $1 \le t < T$ is the expected discounted value of all future dividends plus the redemption value:

$$P_t^* = \frac{V_T}{(1+r)^{T-t}} + \sum_{k=1}^{T-t} \frac{\mathbb{E}[D_k]}{(1+r)^k} \tag{6}$$

While Equation 6 is elegant in that it builds only off of the present values of expected dividend payments and redemption value, it is not immediately clear why $P_t^*$ is constant in our setting. We now show this.

$$P_t^* = \frac{V_T}{(1+r)^{T-t}} + \mathbb{E}[D_k] \sum_{k=1}^{T-t} \frac{1}{(1+r)^k}$$

We may now compute the standard finite geometric sum.

$$P_t^* = \frac{V_T}{(1+r)^{T-t}} + \mathbb{E}[D_k] \cdot \frac{1}{1+r} \left( \frac{1 - (\frac{1}{1+r})^{T-t}}{\frac{r}{1+r}} \right)$$

Manipulating the expression, we obtain the following:

$$P_t^* = \frac{V_T}{(1+r)^{T-t}} + \mathbb{E}[D_k] \cdot \left( \frac{1 - (\frac{1}{1+r})^{T-t}}{r} \right)$$

$$P_t^* = \frac{V_T}{(1+r)^{T-t}} + \frac{\mathbb{E}[D_k]}{r} - \frac{\mathbb{E}[D_k]}{r(1+r)^{T-t}}$$

Recall that $V_T = \$14 = \frac{\$0.7}{0.05} = \frac{\mathbb{E}[D_k]}{r}$. Thus, we can complete our simplification to obtain:

$$P_t^* = \frac{\mathbb{E}[D_k]}{r} \tag{7}$$

Notice that Equation 7 is time independent. Hence, we have shown (1). Further, given the experiment parameters $P_t^* = \$14$, which is exactly the redemption value, demonstrating (2).

## C  EXTENDED RELATED WORKS

Here we provide a more comprehensive overview of prior work at the intersection of LLMs, strategic reasoning, and finance. While the main text highlights only the studies most relevant to our contribution, this appendix details the broader landscape for transparency.

LLMs in strategic environments. Several studies analyze whether LLMs replicate human-like behaviors in canonical game-theory tasks. Brookins & DeBacker (2024) examine GPT-3.5 in the Prisoner's Dilemma, showing tendencies toward cooperation that exceed typical human baselines. Horton (2023) study LLMs in public goods games, highlighting context-dependent altruism. Akata et al. (2023) explore GPT-4 in iterated games, where it adapts between cooperative and competitive strategies, suggesting emergent long-term planning.

Empirical finance applications. A large and rapidly growing literature investigates LLMs as tools for real-world financial prediction and decision-making. Studies include LLM agents for trading and market making Xiao et al. (2024); Li et al. (2024; 2025); Yu et al. (2024b), financial data analysis Yu et al. (2024a); Yun (2024); Liu & Lo (2025); Ding et al. (2024), multimodal integration Zhang et al. (2024b); Yu et al. (2023); Kim (2025), and forecasting tasks Chen (2025); Zhou et al. (2025); Zhang et al. (2024a); Kang et al. (2024); Siddique et al. (2025); Saqur (2024); Bhat & Jain (2024); Kuruvilla & Mythily (2025); Oprea & Bâra (2024); Shi & Hollifield (2024); Cheng & Chin (2024); Zhao et al. (2024). This literature demonstrates the usefulness of LLMs in financial contexts but focuses on exogenous markets and benchmarks.

Experimental and behavioral finance. Our work differs from the above by situating LLMs in endogenous experimental markets, where prices emerge solely from agent interaction. This connects more directly to the tradition of laboratory asset markets Smith et al. (1988); Bostian & Holt (2009), which have long been used to isolate human biases such as herding, overconfidence, and bubble formation. To our knowledge, ours is the first study to apply this paradigm systematically to LLM-agents.

**Evaluation Frameworks for LLMs** Conventional benchmarks for LLMs focus on static environments, emphasizing either knowledge-oriented tasks (e.g., Hendrycks et al. (2021)) or reasoning-oriented challenges (Cobbe et al. (2021)). While these benchmarks provide valuable insights, they often overlook strategic reasoning, a key facet of human intelligence. Recent efforts, such as Chatbot Arena (Zheng et al. (2023)), introduce pairwise comparison frameworks to assess LLM interactions dynamically.

## D  ADDITIONAL TREATMENTS/ROBUSTNESS

### D.1  EXPERIENCED SESSIONS

We tested whether prior exposure changes agent behavior via an experienced-session variant of the mono-market. Each model completed two full 30-round sessions with the standard environment (i.i.d. dividends, 5% cash interest, redemption at 14; prices fully endogenous to submitted orders). In Session 2, the prompt additionally provided the complete market history from Session 1 (prices/volumes/clearing outcomes) and the agent's own PLANS/INSIGHTS files from that session.

Chain-of-thought was enabled in both sessions. Dividend paths were re-sampled; agent sampling (e.g., temperature) was not fixed across runs.

Performance in the "experienced" Session 2 was essentially unchanged relative to Session 1: mean absolute deviation from fundamental value, realized profit, and forecast accuracy remained at similar levels for all models. Figure 9 shows the price paths for the experienced runs. The incidence and magnitude of bubbles did not show a consistent increase or decrease. In short, exposure to the entire prior market, together with perfect recall of that history, did not materially alter trading outcomes. This suggests that substantive learning (if any) may require changing objectives (e.g., fine-tuning/reward shaping) or the environment (e.g., information asymmetries, cross-asset substitution, transaction costs). Note: Grok 2 has been deprecated, and thus we were unable to run this treatment with that model. Mistral-Large had technical errors that will be rerun for camera-ready submission.

Figure 9: Experienced Market Trajectories for different LLMs.

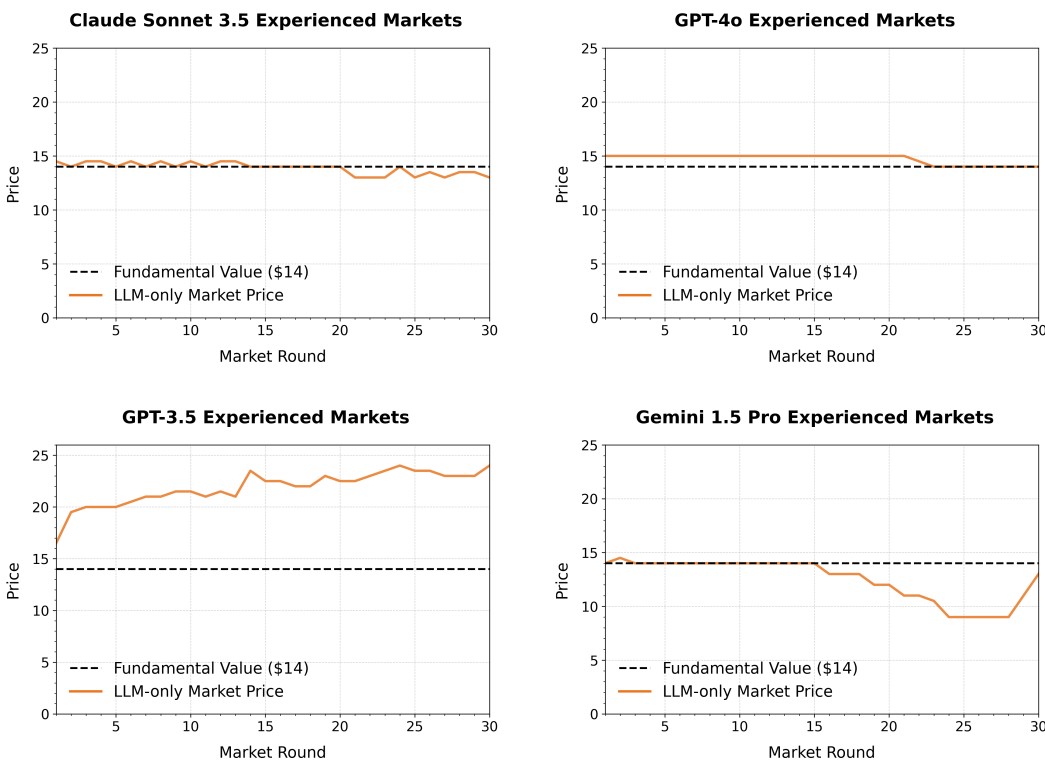

## D.2 RISK ELICITATION

We tested whether inserting a standard risk-elicitation task primes LLM trading. For each model, we ran one mono-market session (30 rounds, i.i.d. dividends, 5% cash interest, redemption value at 14) and added Holt–Laury–style lotteries at the same cadence used in our human studies (interleaved prompts between rounds). The task was exogenous: lottery choices did not affect cash balances, inventory constraints, or market clearing. Prompts otherwise matched the baseline; prices remained fully endogenous to submitted orders.

Adding risk-elicitation did not materially change outcomes relative to the baseline per model, as shown in Figure 10. Mean absolute deviation from fundamental value, bubble incidence/magnitude, realized profit, turnover, and forecast accuracy were all at similar levels; no consistent directional shift appeared in any model class.

In this environment, brief risk-preference queries do not prime LLM agents toward systematically different trading behavior. This aligns with our human-protocol rationale (the task is orthogonal to

trading) and supports the symmetry claim: omitting the lotteries for LLMs in the main experiments does not explain the LLM–human differences we report. Note: Grok 2 has been deprecated, and thus we were unable to run this treatment with that model. Gemini-1.5 had technical errors that will be rerun for camera-ready submission.

Figure 10: Risk-Elicitation Priming Market Trajectories for different LLMs.

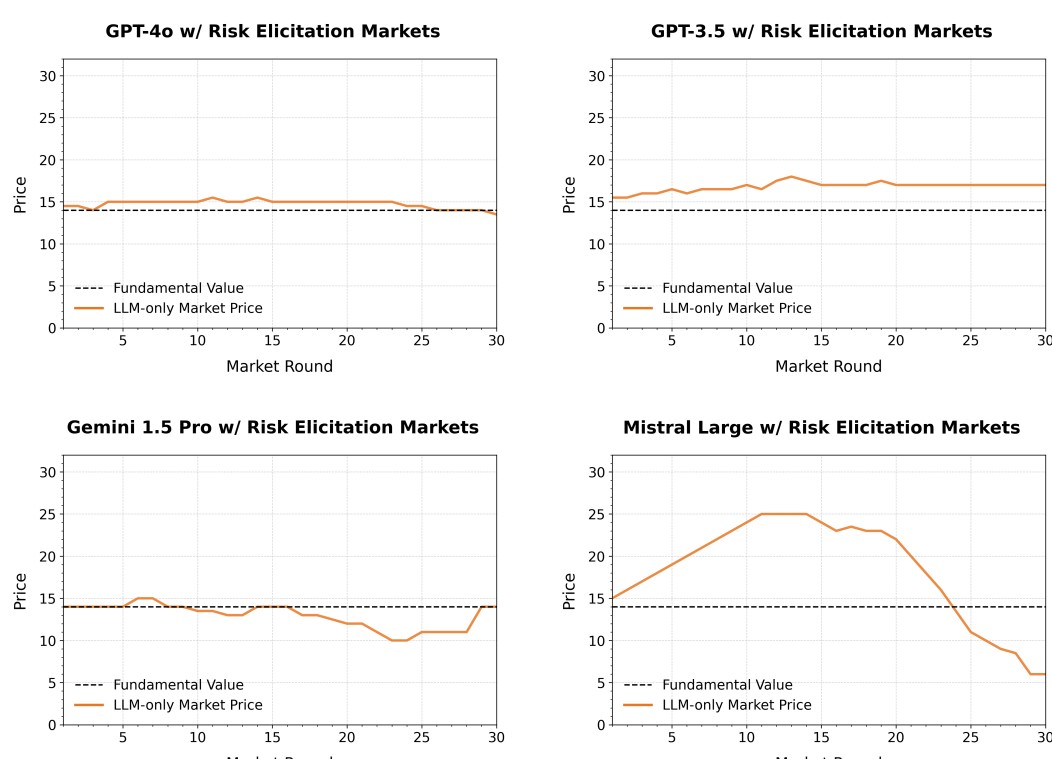

# E ADDITIONAL ANALYSIS

## E.1 HUMAN MARKET TRAJECTORIES

Figure 11: Human Market Trajectories

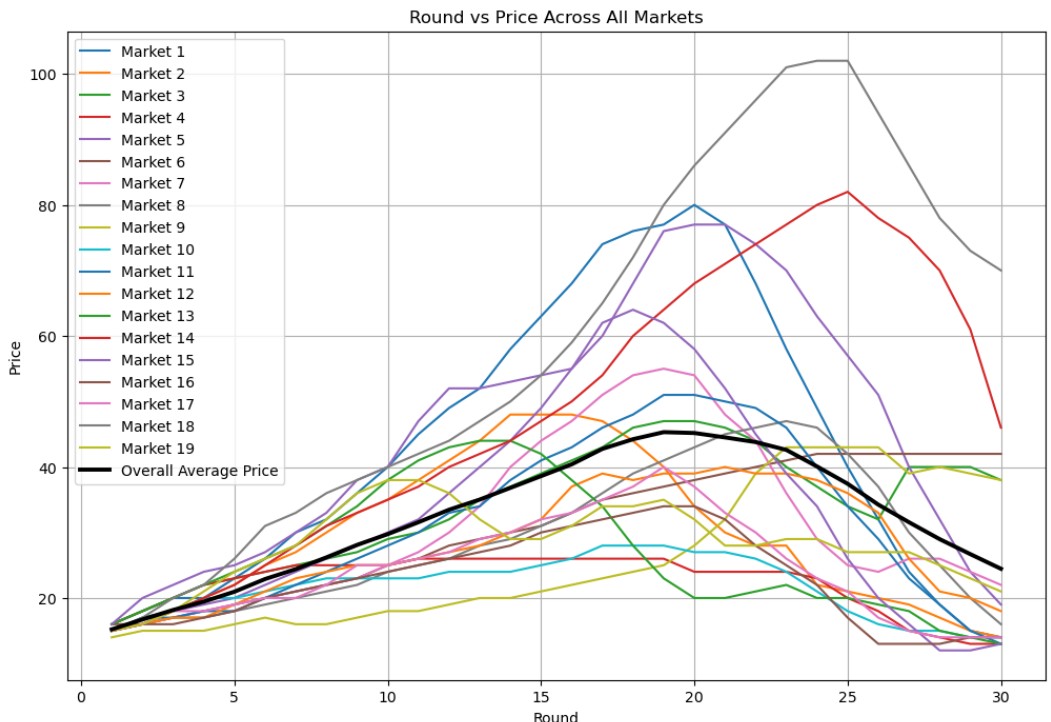

Individual market trajectories from human-only experiments

## E.2 DETAILED INDIVIDUAL MARKET ANALYSIS

- **Gemini 1.5 Pro:** Gemini markets seem to generate a reverse bubble trend, where the asset is initially sold off below fundamental and then bid back up as redemption grows closer (Figure 12). While overall rationality given by the MSE ($\sim$ 3.1) from the fundamental metric is low, the behavior is the exact opposite of what we have seen in the majority of human experiments and indicates that the model likely does not truly understand the task nor model any biases that humans may exhibit.

Figure 12: Gemini 1.5 Pro Markets

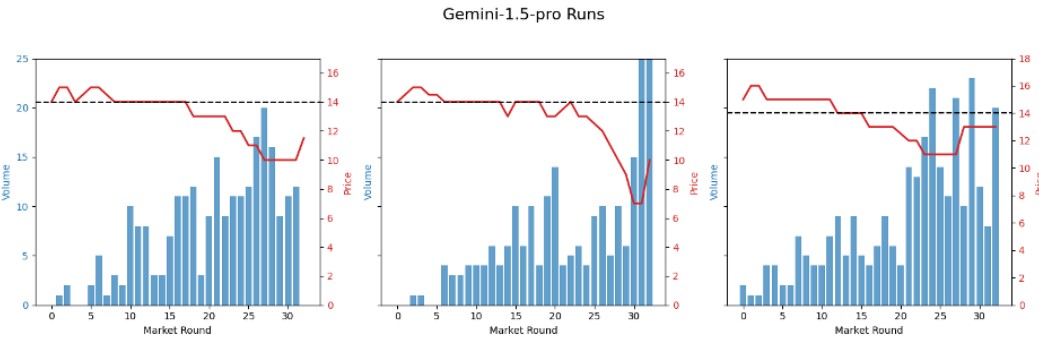

This market seems closest to the Erratic (E) hypothesis. This is also supported by the Pearson correlation coefficient (PCC) as compared to the average human market price of -0.4009, indicating that there are large differences between this model's performance and that of humans.

We can also analyze the uniformity of the market, as you can see in Figure 13, compared to other markets, the traders do seem to have a more varied earning profile. This is also shown in the Portfolio Value variance, which is 45.58, higher than all but one of the other single-model markets.

Figure 13: Gemini 1.5 Pro Portfolio Variance

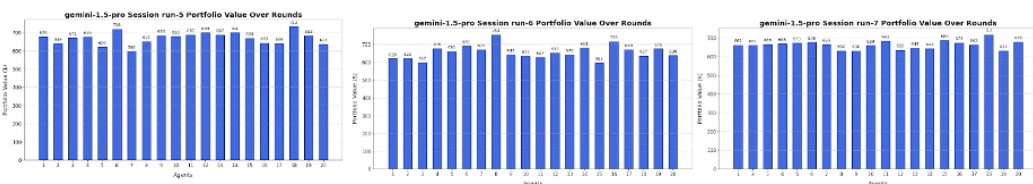

- **Mistral-Large 24.11** Mistral seems to generate markets that appear mostly like the type of bubbles we've seen in our human markets, though the size of the asset bubble is greatly reduced from our median human market and the price crashes below fundamental during the final rounds (Figure 14).

Figure 14: Mistral-Large 24.11 Portfolio Markets

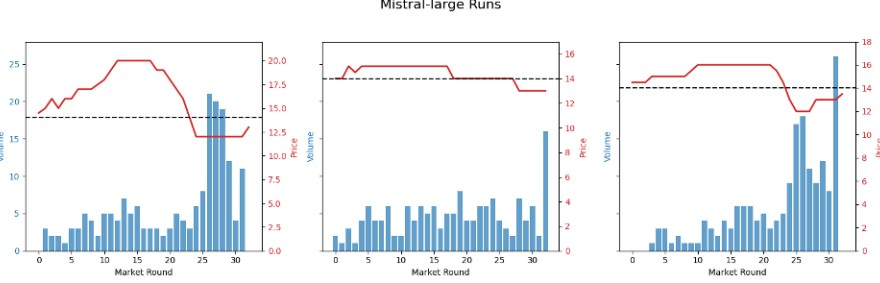

This behavior, while not economically rational (purely rational agents would only buy a price below 14 and would sell any price above 14) does correspond somewhat with the behavior we've seen out of human players, though the Pearson correlation coefficient (PCC) as compared to the average human market price of -0.1118 are large differences. This negative PCC likely pertains to the behavior in these markets after the bubble crashes. This market seems closest to the Human (H) hypothesis.

We can also analyze the uniformity of the market, as can be seen in the below graph (Figure 15), compared to other markets, the traders do seem to have a more varied earning profile. This is also shown in the mean Portfolio Value variance, which is 49.96, the highest of the single-model markets.

Figure 15: Mistral-Large 24.11 Portfolio Variance

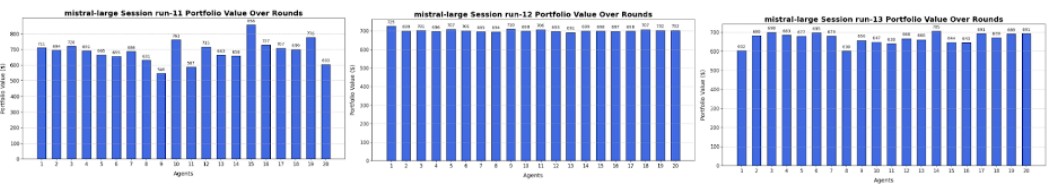

- **Claude-3.5-Sonnet:** Claude markets seem to demonstrate highly 'rational' behavior, the price remains very close to the fundamental value throughout the market (Figure 16). This indicates that the model likely understands the "rational" strategy. This market is the "most" rational based on the MSE metric with the lowest MSE from fundamentals of all the markets ($\sim 0.53$).

Figure 16: Claude-3.5-Sonnet Markets



This market seems closest to R. Which is further supported by the Pearson correlation coefficient (PCC) as compared to the average human market price of being $\sim 0.0013$, showing that there are large differences between this model's performance and that of humans.

We can also analyze the uniformity of the market, as can be seen in the below graph (Figure 17), compared to other markets, the traders do seem to have a more uniform earning profile. This is also shown in the mean Portfolio Value variance, which is 26.15, lower than most of the other single-model markets.

Figure 17: Claude-3.5-Sonnet Portfolio Variance

- **GPT-3.5:** Demonstrates bubble behavior without the crash, where prices rise steadily with no indication of a return to fundamental value (Figure 18). Unlike human markets, trading volume almost disappears toward the end of the experiment, whereas humans generally trade more as the end approaches. This model is the least "rational" by the MSE metric, having a far and away the highest MSE of all the models ($\sim 26.36$).

Figure 18: GPT-3.5 Markets

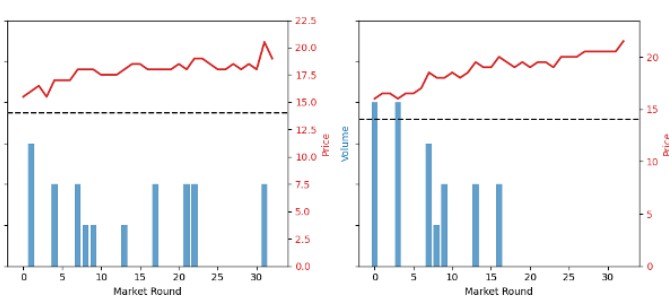

This market seems closest to H, as some bubbles without crashes were observed in human experiments. Interestingly, this model has a relatively high Pearson correlation coefficient (PCC) as compared to the average human market price of $\sim 0.4897$, but it is clear that there are still large differences between this model's performance and that of humans.

The Portfolio Value variance for this market is 30.44 (Figure 19), one of the lower values among single-model markets, indicating generally uniform behavior.

Figure 19: GPT-3.5 Portfolio Variance



- **GPT-4o:** Demonstrates mostly rational behavior, with prices remaining very close to the fundamental value throughout the market (Figure 20). This indicates that the model likely understands the "rational" strategy. This is emphasized quantitatively by the model having the second lowest MSE compared to fundamental at ($\sim 0.79$). This market seems closest to the Rational (R) hypothesis.

Figure 20: GPT-4o Markets

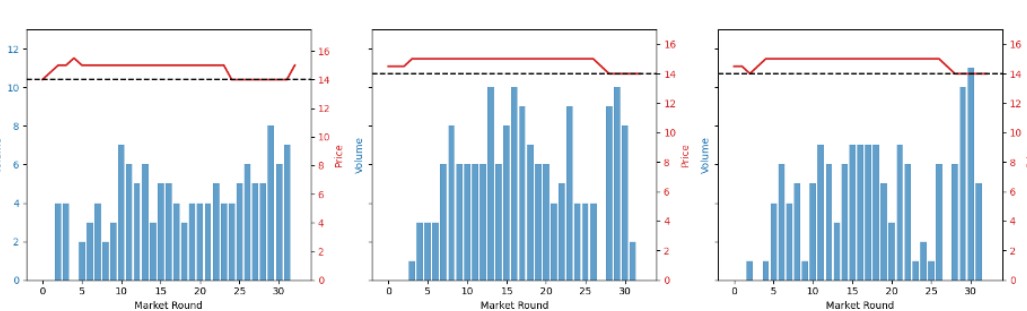

Additionally, unlike human markets, almost all the agents performed very similarly to each other, indicating that there was not a large amount of behavior variance among the agents.

This is also shown in the mean Portfolio Value variance of 22.76 (Figure 21), which is the lowest amongst the single-model markets.

Figure 21: GPT-4o Portfolio Variance



# F    ADDITIONAL NATURAL LANGUAGE PROCESSING INFORMATION

After each round, LLM agents were prompted to provide details of the strategy they were using for their trading behavior, both to enable us to analyze behavior and also to allow the LLMs to develop long-running strategy making use of Chain-of-Thought-like reasoning. In contrast, Human subjects were asked what approach they employed only at the end of the experiment. All comments gathered were preprocessed by converting to lower case, removing punctuation and new line characters, and removing stop words. (English stop words from Python's nltk library) For the human subjects, this part was optional. Even so, we received 442 responses from 514 subjects (85.99%), averaging 18.95 words each. The 620 individual LLM agents generated 18,060 responses, averaging 64.75 words each.

First, we looked at the most frequently used words in each group of participants (humans and LLMs). Of the top six words for each group, five are the same, though in a different order ("sell", "buy", "stock", "price", and "market") (Table 6). The top fifteen most popular words for the human subject include "low", "high", and "early", which do not appear among the top fifteen words used by the LLMs, while the words "continue", "monitor", "closely", and "buyback" are among the most popular words for the LLMs. (Popular words are highlighted in word clouds shown in Figure 22.) This suggests that the human traders might follow a "buy low, sell high" strategy in an attempt to beat the market, while the LLM agents take a longer investment approach, keeping the fundamental value of the asset in mind.

To confirm this idea, we ran a Latent Dirichlet Allocation (LDA) Blei et al. (2003) on all strategies combined. We used the Python gensim library Řehůřek & Sojka (2010) to assign the strategies to one of two topics. The topic identified by the words *orders, sell, buy, shares, stock* denoted strategies that implement a "buy low; sell high" practice. We noticed that 89.3% of human strategies were assigned this topic by the LDA analysis, while only 36.6% of the LLM strategies. A t-test of this difference is highly significant (p < .001). The other topic was identified by the words *stock, adjust, buyback, closely, current* and suggests a more measured and conservative approach. The LDA analysis assigned 63.4% of the LLM strategies (10.7% of humans) to these topics. Figure 23 shows this assignment.

## F.1    INDIVIDUAL TOKEN COUNTS

### F.1.1    HOW THE MODELS' LANGUAGE REVEALS DISTINCT TRADING PLAYBOOKS

**Method.**    For every agent, we concatenate the round-by-round `Plan` and `Insight` fields, convert them to lowercase, strip punctuation, and tokenize.

Four interpretable features are then computed:

1. **Fundamental token rate** – occurrences of *fundamental, intrinsic, value, mean-revert, FV, 14, discounted cash-flow* (14 keywords).
2. **Speculative token rate** – *pump, breakout, leverage, surge, moon, momentum, Fomo* (18 keywords).
3. **Trade-verb density** – *buy, sell, long, short, liquidate*.

Table 6: Most popular words used by human participants and by the LLM Agents

| Rank | Human | | | LLM-Agent | | |
| --- | Word | Count | Freq | Word | Count | Freq |
| 1 | **sell** | 230 | 5.60% | **price** | 88707 | 6.04% |
| 2 | **buy** | 202 | 4.92% | **market** | 63445 | 4.32% |
| 3 | **stock** | 177 | 4.31% | **buy** | 38996 | 2.66% |
| 4 | **price** | 141 | 3.43% | **sell** | 38141 | 2.60% |
| 5 | tried | 129 | 3.14% | orders | 31335 | 2.14% |
| 6 | **market** | 79 | 1.92% | round | 28385 | 1.93% |
| 7 | low | 75 | 1.83% | **stock** | 28130 | 1.92% |
| 8 | high | 70 | 1.70% | **cash** | 26469 | 1.80% |
| 9 | would | 57 | 1.39% | continue | 23318 | 1.59% |
| 10 | interest | 53 | 1.29% | adjust | 19250 | 1.31% |
| 11 | **shares** | 51 | 1.24% | value | 18669 | 1.27% |
| 12 | strategy | 48 | 1.17% | **shares** | 18476 | 1.26% |
| 13 | **cash** | 44 | 1.07% | buyback | 17555 | 1.20% |
| 14 | early | 41 | 1.00% | monitor | 16582 | 1.13% |
| 15 | much | 41 | 1.00% | current | 14595 | 0.99% |

Common words are in bold text. The Count columns are the number of times a word is used by each group (humans and LLMs). Frequency is the count divided by the number of individual words used by that group.

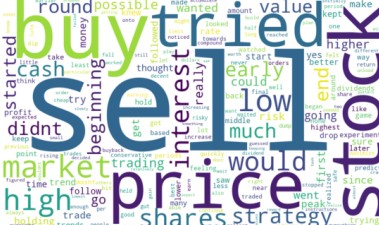

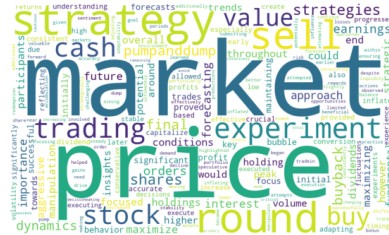

(a) Human Strategies

(b) LLM Strategies

Figure 22: Word cloud representation of the most commonly used words in the strategies for human and LLM agents.

4. **Risk / Profit ratio** – risk-related tokens divided by profit-related tokens (*risk, volatility, drawdown* vs. *profit, gain, return*).

Counts are normalized per 1,000 tokens

Table 7: Language–based strategy fingerprints (plans+insights).

| Model | Fund. /1000 | Spec. /1000 | Trade verbs/1000 | Risk /Profit |
| --- | --- | --- | --- | --- |
| Claude-3.5 Sonnet | **61.9** | 4.3 | 21.7 | 1.98 |
| GPT-4o | 26.0 | 4.1 | **31.5** | **0.90** |
| Gemini-1.5-Pro | 40.1 | 7.0 | 34.8 | 1.01 |
| Grok-2 | 29.9 | 5.0 | 17.8 | 2.30 |
| Mistral-Large | 25.7 | 6.5 | 34.2 | 1.12 |
| GPT-3.5 | 19.6 | **10.1** | **3.8** | **7.55** |

**Findings.**

Figure 23: Percentage of human and LLM strategies assigned to one of two topics.

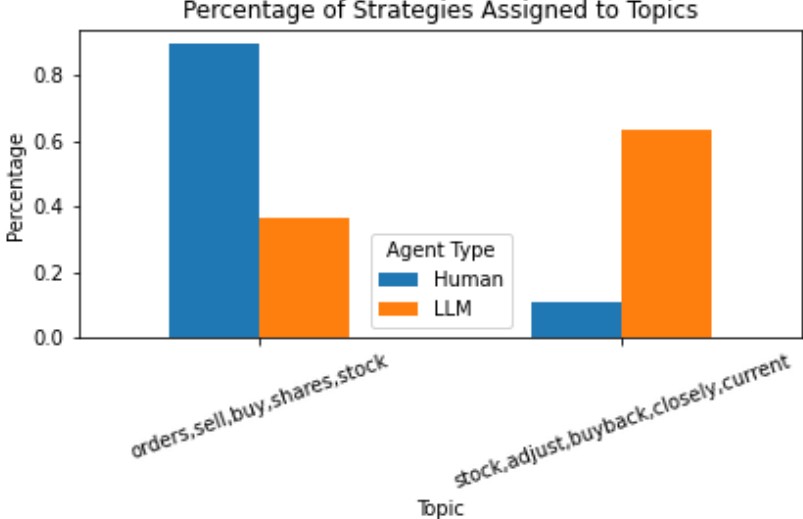

The topics were generated and assigned by Latent Dirichlet Allocation. The topic *price, round, orders, sell, buy* indicates strategies that attempt to beat the market (buy low; sell high). While the *price market continue, stock, adjust* topic suggests a more measured and long-term strategy.

- **Value-anchored rationalists.** Claude-3.5 (62 fundamental tokens) and GPT-4o (lowest Risk/Profit) talk most about intrinsic value and trade in balanced lots, keeping price within $1–2 of FV.
- **Momentum bubble–builders.** Grok-2 and Mistral-Large use twice the speculative language of Claude and issue 18–34 trade verbs per 1,000 → collective buying pushes price up to $20–22 before a late crash.
- **Risk-obsessed inflator.** GPT-3.5 has the leanest trade language but a Risk/Profit ratio ≈8 ×. Agents buy cautiously, hold, and never realize gains, so the market inflates to $18.8 yet never panics.
- **Net-seller undervaluation.** Gemini's speculative share is moderate, but its sell verbs outnumber buys (net-buy = –0.19), driving the only "reverse bubble" that trades *below* FV.

A simple multinomial logistic regression on these four features classifies the six models with 83% 5-fold CV accuracy (Appendix F), confirming that each family follows a linguistically—and behaviorally—distinct playbook.

### F.2 SENTIMENT ANALYSIS

Sentiment analysis (or opinion mining) is a natural language processing technique that measures the emotional tone of a passage of text. The aim is to assign a score that is either negative or positive to a passage to express whether it conveys a negative or positive affect. We use `nltk` (Bird et al., 2009) to perform our analysis. We did not word stem or remove stopwords after Pang et al. (2002).

We ran a sentiment analysis on the strategies and found that human and LLM agents produce a similar distribution of sentiment (Figure 24). This similarity might be due to the relatively banal nature of the subject. However, the human subjects did produce a slightly wider variance of sentiment ($\sigma_h = 0.167 > \sigma_b = 0.131$).

The strategies of the LLM agents were collected after each round, whereas the human subjects were asked to provide a strategy once after the experiment. This difference allows us to examine the changes in sentiment over time and market conditions. We noticed that sentiment is significantly different when the market price is below the fundamental value (Figure 25). We define mispricing

Figure 24: Distributions of sentiment for human and LLM agents

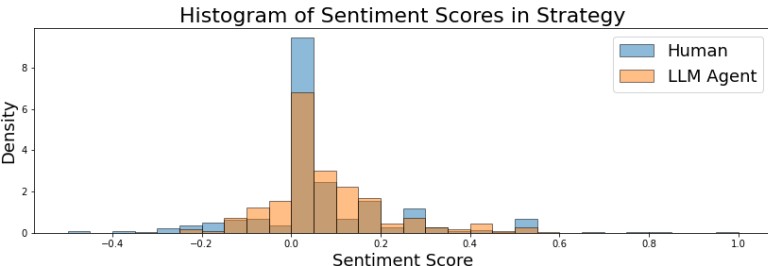

Average human sentiment was $\mu_h = 0.061$ ($\sigma_h = 0.167$). Average LLM sentiment was $\mu_b = 0.072$ ($\sigma_b = 0.131$)

. A t-test revealed no significant difference (t=-1.842; p=0.063).

|  | Sentiment x 100 |
| --- | --- |
| Mispricing | 0.294** |
|  | (0.0674) |
| Round | -0.136** |
|  | (0.0275) |
| Constant | 8.684*** |
|  | (0.358) |
| Observations | 18600 |

Standard errors in parentheses
$^{*}\ p < 0.05$, $^{**}\ p < 0.01$, $^{***}\ p < 0.001$

Table 8: Fixed effects regression of sentiment score on round number and level of mispricing. Standard errors are clustered on LLM and shown in parentheses. Sentiment was multiplied by 100 to reduce the significant digits in the estimates.

as $M_{sr} = MP_{sr} - FV_s$ where $M_{sr}$ is the mispricing for session $s$ in round $r$, $MP_{sr}$ is the market price in session $s$ round $r$, and $FV_s$ is the fundament value for session $s$ (which is 14 for all sessions). We ran a fixed effects regression of sentiment scores on round number and mispricing with standard errors clustered on the LLM (table 8). The results show that the models do tend to adjust the tone of their strategy based on market conditions, even when controlling for the specific LLMs.

# G  RATIONALITY PROPERTIES OF INDIVIDUAL LLM FORECASTS

In addition to analyzing the pricing behavior of LLMs, we were also interested in how these models generate forecasts for future price periods. We captured each model's forecasts, which allowed us to assess not only the rationality of their forecasts, whether they align with fundamental valuation principles, but also whether the LLMs themselves behave rationally in response to their expectations. We compare this analysis to that of our human traders to understand whether LLMs, in the Single-LLM market setting, make similar or different forecasting mistakes compared to their human counterparts. The mathematical formulation for calculating the mean and median forecasts for any given session is provided in the appendix (Appendix G.3).

## G.1  FORECASTING FRAMEWORK

Fix a session (we will suppress session notation). For each period $t$, participants provide forecasts before the price revelation. The forecast made in period $t$, denoted $f_{i,t}^t$, predicts the price $P_t$. The forecast for future periods, made in period $t$, is denoted as $f_{i,t}^{(t+h)}$, where $h \in \{0, 2, 5, 10\}$ is the prediction horizon (or how far into the future agents are forecasting).

Figure 25: Sentiment scores are plotted against the mispricing of the market compared to the fundamental value of \$14

(Mispricing = Market Price - FV) The shaded region represents the 95% confidence interval.

The forecast error for an individual $i$ in period $t$, forecasting $P_{t+h}$, is given by: $E_{i,t}^h = P_{t+h} - f_{i,t}^{(t+h)}$ A positive value indicates that the agent underforecasted the price, while a negative value indicates an overforecast. In the appendix, Table 10 presents the full descriptive statistics of prices and forecasts, while Table 11 reports the forecast errors for each model and for our human participants.

For next-round forecasts, the Mistral-Large model achieves the smallest average error (0.00), highlighting its most rational performance. GPT-4o follows with an error of 0.04, Claude-3.5-Sonnet with 0.09, and Gemini-1.5-Pro with 0.14. Grok-2 and GPT-3.5 record errors of –0.31 and –1.54, respectively. By comparison, human participants exhibit a one-round forecast error of approximately 1.67. Thus, all AI models outperform human forecasters in terms of forecast accuracy.

## G.2 DISTRIBUTION OF FORECAST ERRORS

We investigate the distribution of forecast errors for each AI model across each horizon, grouped by experimental rounds. Figure 26 demonstrates that these distributions vary both by model and by prediction horizon.

At the model level, all the models exhibit more tightly centered error distributions around 0 when compared to human markets. While GPT-3.5's errors are the most dispersed, it has notably more concentrated distributions than human participants. GPT-3.5's distribution also shows a leftward shift, most notably at the five- and ten-period horizons, indicating an increasing underforecast bias over successive rounds, a pattern that somewhat parallels the trend among human forecasters. The other models do not show this shift.

Next, we compute the mean forecast error for each round and horizon across all AI models and human participants. Figure 5 shows how this average error evolves over successive rounds. Among the models, Claude-3.5-Sonnet performs best—its mean error remains closest to zero at every horizon. GPT-4o initially overforecasts but steadily converges toward zero in later rounds, reflecting increasing accuracy. By contrast, GPT-3.5 consistently overforecasts as rounds progress, making it the least accurate model. Notably, none of the AI models replicate the human pattern of early underforecasting and then subsequent overforecasting in later rounds.

### G.2.1 UNBIASEDNESS

At the individual agent level, we test whether the mean forecast error $E_{i,t}^h$ is zero across $t$ for each horizon $h$. This yields four test statistics per agent, corresponding to the four horizons. For each case, we conduct $t$-tests to determine whether the mean forecast error is significantly different from zero, i.e., whether it is biased. For each horizon, we then count the number of agents whose mean forecast error is not significantly different from zero, indicating unbiased forecasts, and compute

Figure 26: Comparison of forecast errors across rounds for human participants and AI models

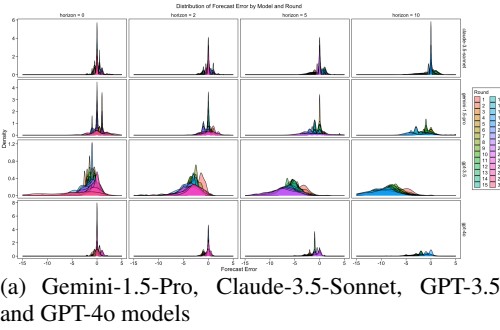

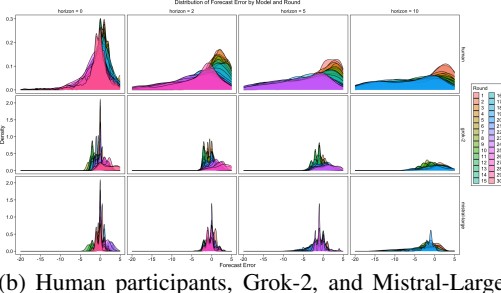

(a) Gemini-1.5-Pro, Claude-3.5-Sonnet, GPT-3.5, and GPT-4o models

(b) Human participants, Grok-2, and Mistral-Large models

the percentage of unbiased agents for each model. The first block of Table 9 presents, for each model and horizon, the percentage of agents whose forecasts are unbiased. The corresponding results are also shown in Figure 31. Among the models, GPT-3.5's forecasts are the most biased, as all forecast errors for the four horizons differ from 0. Claude-3.5-Sonnet, Gemini-1.5-Pro perform most similarly to humans; the bias measure decreases as the horizon increases. GPT-4o performs best at horizon = 0, indicating the most rational forecasting and the least bias.

Table 9: Rationality test results

| Test | Horizon | Human | Claude-3.5 Sonnet | Gemini-1.5 Pro | GPT-3.5 | GPT-4o | Grok-2 | Mistral-Large |
|---|---|---|---|---|---|---|---|---|
| Unbiasedness | 0 | 0.619 | 0.833 | 0.778 | 0 | 1.000 | 0.657 | 0.972 |
| | 2 | 0.623 | 0.389 | 0.583 | 0 | 0.597 | 0.554 | 0.528 |
| | 5 | 0.570 | 0.306 | 0.194 | 0 | 0.000 | 0.516 | 0.139 |
| | 10 | 0.438 | 0.292 | 0.069 | 0 | 0.000 | 0.547 | 0.111 |
| Zero autocorrelation | 0 | 0.420 | 0.069 | 0.208 | 0.819 | 0.458 | 0.031 | 0.194 |
| | 2 | 0.092 | 0.236 | 0.208 | 0.847 | 0.648 | 0.016 | 0.208 |
| | 5 | 0.068 | 0.000 | 0.083 | 0.806 | 0.403 | 0.062 | 0.139 |
| | 10 | 0.086 | 0.028 | 0.208 | 0.861 | 0.486 | 0.109 | 0.097 |
| Forecast error uncorrelated with forecast levels | 0 | 0.625 | 0.028 | 0.389 | 0 | 0.056 | 0.438 | 0.403 |
| | 2 | 0.349 | 0.308 | 0.431 | 0 | 0.069 | 0.203 | 0.472 |
| | 5 | 0.145 | 0.175 | 0.586 | 0 | 0.000 | 0.141 | 0.306 |
| | 10 | 0.117 | 0.123 | 0.643 | 0 | 0.000 | 0.125 | 0.250 |

### G.2.2 ZERO AUTOCORRELATION

To test for zero autocorrelation, we examine whether forecast errors from period $t$ should be independent of forecast errors in subsequent periods. The guiding principle is that future forecast errors should not depend on the information available at the time the forecast was made.

To test this, we estimate the regression: $E_{t,0} = \beta_0 + \beta_1 E_{t-1,0} + \epsilon_t$, where the hypothesis is $\beta_1 = 0$. This zero-autocorrelation condition extends similarly to longer forecast horizons.

We divide the data at horizon $h = 0$ by agent and use ordinary least squares regression to estimate $\beta_1$ for each agent at different horizons. For each model and horizon, we then count the number of agents whose $\beta_1$ is not significantly different from zero—indicating zero autocorrelation and thus rationality—and calculate the corresponding percentage.

The second block of Table 9 presents the results of the autocorrelation test (see also Figure 28). We see some heterogeneity among the models; GPT-3.5 appears to be the most rational, with the highest percentage of agents showing no autocorrelation. In contrast, Claude-3.5-Sonnet and Grok-2 exhibit the strongest autocorrelation, with the lowest percentage of agents passing the test. Humans show autocorrelation when the horizon is 0, a pattern similar to GPT-4o (humans: 42%, GPT-4o: 45.8%).

### G.2.3 CORRELATION BETWEEN FORECAST ERRORS AND FORECASTS

Forecast errors, $P_{t+h} - f_{i,t}^{(t+h)}$, should be uncorrelated with forecasts: $\text{cor}(P_{t+h} - f_{i,t}^{(t+h)}, f_{i,t}^{(t+h)}) = 0$ This principle is grounded in two key intuitions. First, the current forecast is part of an agent's information set. If any variable within that set can predict future forecast errors, a rational agent should incorporate it to reduce those errors. Second, a negative correlation between forecast errors and forecasts suggests a systematic bias: when an agent is optimistic, making a high forecast, they tend to overestimate the actual outcome, resulting in a negative forecast error.

To test this, we compute these correlations separately for each forecast horizon and agent by estimating the regression: $E_{i,t}^h = \beta_0 + \beta_1 f_{i,t}^{(t+h)} + \epsilon_{i,t}^h$

For the four values of $h$, we count the number of agents whose $\beta_1$ is not significantly different from zero for each model and horizon, indicating no correlation between forecast errors and forecasts, which is consistent with rationality.

The third block of Table 9 reports the results of the correlation test across different horizons (see also Figure 33). When the horizon is 0, humans perform the best, followed by Grok-2. In contrast, GPT-3.5 and GPT-4o perform the worst, as their forecast errors are consistently correlated with their forecasts across nearly all horizons.

In summary, LLMs generally produce more stable and centered forecast errors than humans. GPT-3.5 shows high bias and strong forecast-error correlation but low autocorrelation. Claude-3.5-Sonnet and Gemini-1.5-Pro are closest to human behavior, while GPT-4o performs well at short horizons. Each model has its strengths and weaknesses, but no model fully replicates human behavior across all tests.

Figure 27: Percentage of Forecasts Without Autocorrelation Bias for Each Model Across Horizons

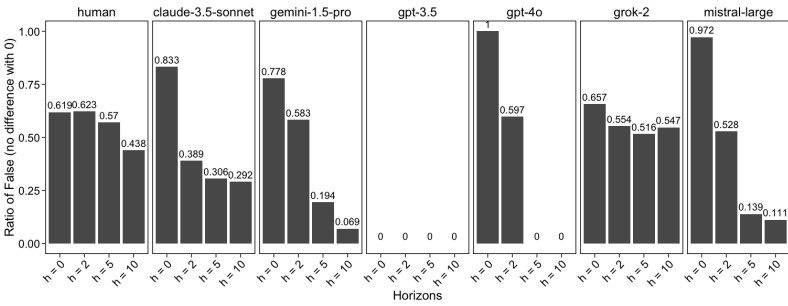

Figure 28: Percentage of Unbiased Forecasts for Each Model Across Horizons

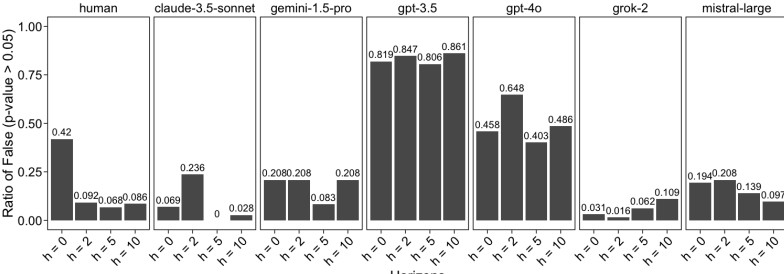

Table 10: Descriptive statistics for Price and Forecasts by source.

| Source | Variable | Mean | SD | Min | Max | Q1 | Q3 |
|---|---|---|---|---|---|---|---|
| claude-3.5-sonnet | Price | 14.65 | 1.00 | 13.00 | 20.00 | 14.00 | 15.00 |
| | Forecast current | 14.67 | 0.98 | 13.00 | 20.00 | 14.00 | 15.00 |
| | Forecast 2 rounds later | 14.78 | 1.12 | 14.00 | 21.00 | 14.00 | 15.00 |
| | Forecast 5 rounds later | 14.90 | 1.49 | 14.00 | 22.00 | 14.00 | 16.00 |
| | Forecast 10 rounds later | 15.13 | 2.04 | 14.00 | 25.00 | 14.00 | 16.00 |
| gemini-1.5-pro | Price | 13.43 | 2.02 | 7.00 | 20.00 | 13.00 | 14.50 |
| | Forecast current | 13.68 | 1.91 | 6.00 | 21.00 | 13.00 | 15.00 |
| | Forecast 2 rounds later | 13.87 | 1.77 | 7.00 | 22.00 | 14.00 | 15.00 |
| | Forecast 5 rounds later | 14.26 | 1.48 | 8.00 | 23.00 | 14.00 | 15.00 |
| | Forecast 10 rounds later | 14.55 | 1.11 | 9.00 | 22.00 | 14.00 | 15.00 |
| gpt-3.5 | Price | 16.48 | 0.92 | 14.00 | 20.00 | 16.00 | 17.00 |
| | Forecast current | 17.98 | 2.15 | 14.00 | 35.00 | 17.00 | 18.00 |
| | Forecast 2 rounds later | 19.92 | 2.06 | 14.00 | 38.00 | 19.00 | 21.00 |
| | Forecast 5 rounds later | 22.69 | 2.42 | 15.00 | 45.00 | 21.00 | 24.00 |
| | Forecast 10 rounds later | 25.28 | 2.76 | 17.00 | 55.00 | 24.00 | 27.00 |
| gpt-4o | Price | 14.94 | 0.87 | 14.00 | 20.00 | 15.00 | 15.00 |
| | Forecast current | 14.95 | 0.91 | 14.00 | 21.00 | 15.00 | 15.00 |
| | Forecast 2 rounds later | 15.20 | 1.10 | 13.00 | 22.00 | 15.00 | 15.00 |
| | Forecast 5 rounds later | 15.98 | 1.45 | 14.00 | 25.00 | 15.00 | 16.00 |
| | Forecast 10 rounds later | 17.06 | 2.10 | 14.00 | 28.00 | 16.00 | 18.00 |
| grok-2 | Price | 18.17 | 2.13 | 14.00 | 22.00 | 16.00 | 20.00 |
| | Forecast current | 18.15 | 2.45 | 14.00 | 23.00 | 16.00 | 20.00 |
| | Forecast 2 rounds later | 18.39 | 2.64 | 14.00 | 24.00 | 16.00 | 20.00 |
| | Forecast 5 rounds later | 18.90 | 2.76 | 14.00 | 26.00 | 17.00 | 21.00 |
| | Forecast 10 rounds later | 19.07 | 3.02 | 14.00 | 28.00 | 17.00 | 21.75 |
| mistral-large | Price | 15.17 | 2.09 | 12.00 | 20.00 | 14.00 | 16.00 |
| | Forecast current | 15.34 | 2.21 | 10.00 | 22.00 | 14.00 | 16.00 |
| | Forecast 2 rounds later | 15.77 | 2.45 | 9.00 | 23.00 | 14.00 | 17.00 |
| | Forecast 5 rounds later | 16.60 | 2.75 | 9.00 | 25.00 | 14.00 | 18.00 |
| | Forecast 10 rounds later | 17.76 | 3.23 | 14.00 | 27.00 | 15.00 | 19.00 |
| Human | Price | 31.31 | 16.42 | 12.00 | 102.00 | 19.00 | 40.00 |
| | Forecast current | 15.50 | 8.66 | 1.00 | 30.00 | 8.00 | 23.00 |
| | Forecast 2 rounds later | 16.50 | 8.08 | 3.00 | 30.00 | 9.75 | 23.25 |
| | Forecast 5 rounds later | 18.00 | 7.21 | 6.00 | 30.00 | 12.00 | 24.00 |
| | Forecast 10 rounds later | 20.50 | 5.77 | 11.00 | 30.00 | 15.75 | 25.25 |

Figure 29: Results Showing the Correlation Between Forecast Errors and Forecasts

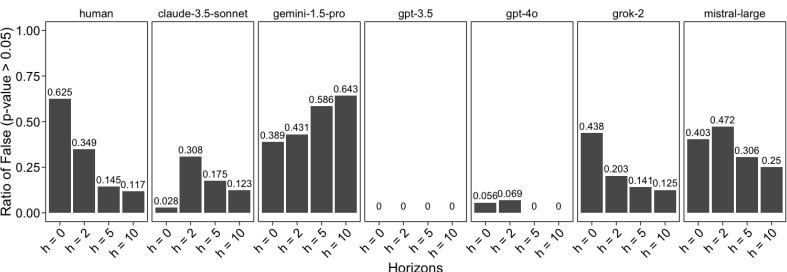

Table 11: Descriptive statistics for Forecast errors by source.

| Source | Variable | Mean | SD | Min | Max | Q1 | Q3 |
|---|---|---|---|---|---|---|---|
| claude-3.5-sonnet | Forecast error current | 0.09 | 0.49 | -2.00 | 3.00 | 0.00 | 0.00 |
| | Forecast error 2 rounds | -0.01 | 0.68 | -2.00 | 3.00 | 0.00 | 0.00 |
| | Forecast error 5 rounds | -0.12 | 1.10 | -4.00 | 4.00 | -1.00 | 1.00 |
| | Forecast error 10 rounds | -0.38 | 1.52 | -7.00 | 3.00 | -1.00 | 1.00 |
| gemini-1.5-pro | Forecast error current | 0.14 | 0.88 | -5.00 | 6.00 | 0.00 | 0.50 |
| | Forecast error 2 rounds | -0.12 | 0.84 | -4.00 | 4.00 | -1.00 | 0.00 |
| | Forecast error 5 rounds | -0.64 | 1.36 | -5.00 | 5.00 | -1.00 | 0.00 |
| | Forecast error 10 rounds | -1.23 | 1.69 | -6.00 | 6.00 | -2.00 | 0.00 |
| gpt-3.5 | Forecast error current | -1.54 | 1.92 | -16.00 | 2.00 | -2.00 | -0.50 |
| | Forecast error 2 rounds | -3.42 | 1.73 | -18.00 | 0.50 | -4.00 | -2.50 |
| | Forecast error 5 rounds | -6.12 | 2.24 | -26.00 | 1.50 | -7.00 | -5.00 |
| | Forecast error 10 rounds | -8.64 | 2.68 | -39.00 | 0.00 | -10.00 | -7.00 |
| gpt-4o | Forecast error current | 0.04 | 0.41 | -2.00 | 2.50 | 0.00 | 0.00 |
| | Forecast error 2 rounds | -0.19 | 0.46 | -2.00 | 2.00 | 0.00 | 0.00 |
| | Forecast error 5 rounds | -0.94 | 0.94 | -6.00 | 1.00 | -1.00 | 0.00 |
| | Forecast error 10 rounds | -2.05 | 1.68 | -11.50 | 2.00 | -3.00 | -1.00 |
| grok-2 | Forecast error current | -0.31 | 1.37 | -3.00 | 6.00 | -1.00 | 0.00 |
| | Forecast error 2 rounds | -0.31 | 1.49 | -3.00 | 6.00 | -1.50 | 0.00 |
| | Forecast error 5 rounds | -0.42 | 1.67 | -5.00 | 5.00 | -2.00 | 1.00 |
| | Forecast error 10 rounds | 0.00 | 2.53 | -7.50 | 7.00 | -2.00 | 2.00 |
| mistral-large | Forecast error current | 0.00 | 1.04 | -4.00 | 4.00 | 0.00 | 0.00 |
| | Forecast error 2 rounds | -0.36 | 0.95 | -4.00 | 3.00 | -1.00 | 0.00 |
| | Forecast error 5 rounds | -1.14 | 1.50 | -9.00 | 4.00 | -2.00 | 0.00 |
| | Forecast error 10 rounds | -2.38 | 2.96 | -14.00 | 6.00 | -3.00 | 0.00 |
| Human | Forecast error current | 1.67 | 8.70 | -84.00 | 92.00 | -1.00 | 2.00 |
| | Forecast error 2 rounds | 1.69 | 11.12 | -79.00 | 95.00 | -3.00 | 6.00 |
| | Forecast error 5 rounds | -0.10 | 17.49 | -149.00 | 96.00 | -7.00 | 9.00 |
| | Forecast error 10 rounds | -0.53 | 25.44 | -178.00 | 96.00 | -12.00 | 15.00 |

## G.3 Additional Forecast Information

### G.3.1 Mean and Median Forecasts

The mean forecast for session $s$ at period $t + h$ is:

$$m_t^{(t+h)} = \frac{\sum_{i=1}^{N_s} f_{i,t}^{(t+h)}}{N_s}$$

Where $N_s$ is the number of participants in session $s$.

The median forecast is calculated as follows: Arrange the values of $f_{t,t+h}^{(i)}$ in ascending order:

$$f_{t,t+h}^{(1)}, f_{t,t+h}^{(2)}, \ldots, f_{t,t+h}^{(N_s)}$$

where

$$f_{t,t+h}^{(1)} \leq f_{t,t+h}^{(2)} \leq \cdots \leq f_{t,t+h}^{(N_s)}$$

If $N_s$ is odd:

$$\text{Median} = f_{t,t+h}^{\left(\frac{N_s+1}{2}\right)}$$

If $N_s$ is even:

$$\text{Median} = \frac{f_{t,t+h}^{(N_s/2)} + f_{t,t+h}^{(N_s/2+1)}}{2}$$

### G.3.2 ADDITIONAL DISTRIBUTION OF FORECAST ERRORS GRAPH

Figure 30: Additional Forecast Error Distribution Graph

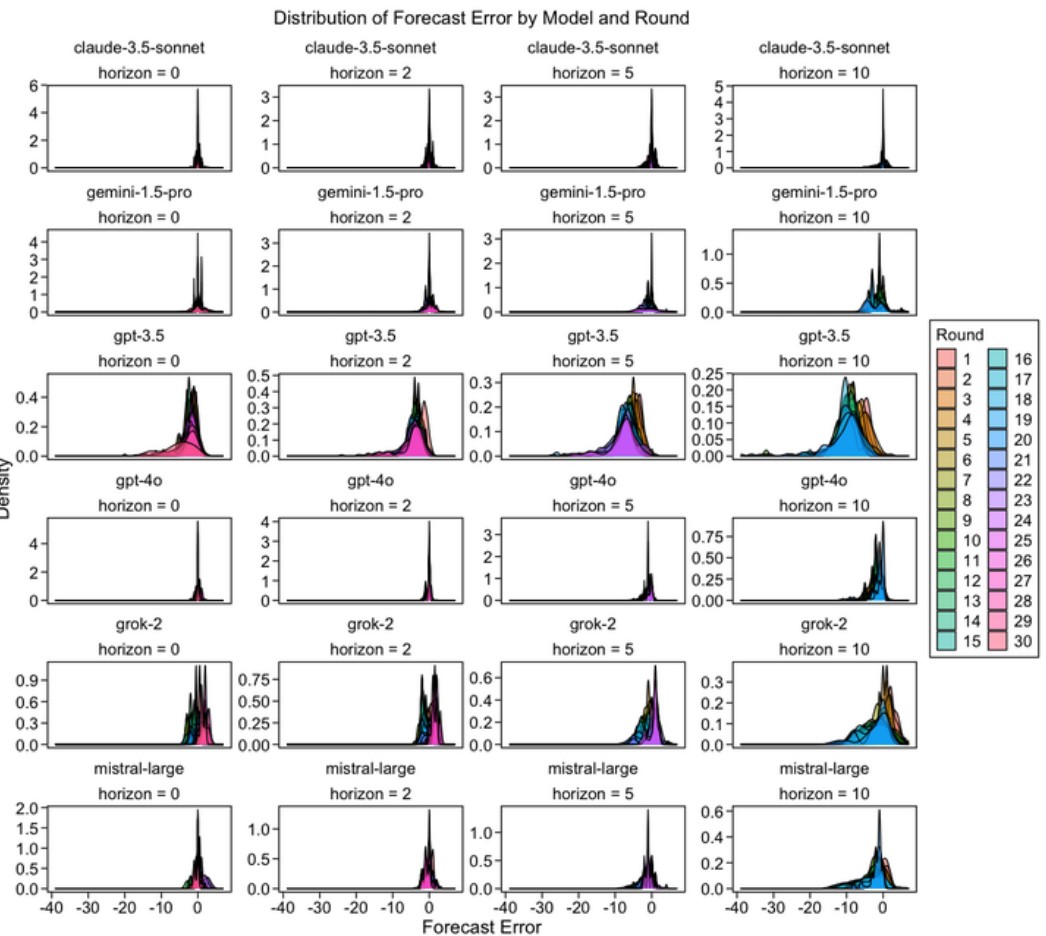

### G.4 MEDIAN FORECAST ERROR

We calculate the median forecast error for each round across different horizons. Figure 5 in the main text illustrates how the median forecast error evolves over time for each horizon. Claude-3.5-Sonnet performs the best, as its mean forecast for each horizon remains closest to 0 compared to other models. GPT-4o shows a trend of initially overforecasting prices before gradually shifting toward 0 in later rounds, indicating a movement toward greater accuracy. In contrast, GPT-3.5 performs the worst, consistently overforecasting prices as the rounds progress.

### G.4.1 UNBIASEDNESS

At the individual agent level, we test whether the mean forecast error $E_{i,t}^h$ is zero across $t$ for each horizon $h$. This provides four test statistics for each value of $h$. We conduct $t$-tests to determine whether the mean forecast error is statistically different from zero.

Specifically, we test whether the mean forecast error for each model and horizon is significantly different from zero. We use the t test at agent level for four horizons and get the p value. Then we count the number of agents whose mean forecast error is significantly different from 0 at each horizon, and calculate the percentage of each model. Figure 31 presents the percentage of no difference with 0 of each model at each horizon. Among the models, gpt-3.5 performs worst, as all forecast error for four horizons are all different from 0, Claude-3.5-Sonnet, Gemini-1.5-Pro performs most similarly

to humans, the ratio gets lower as horizon increase. GPT-4o performs best at horizon = 0, indicating rational forecasting.

Figure 31: The percentage of no autocorrelation bias of each model at each horizon

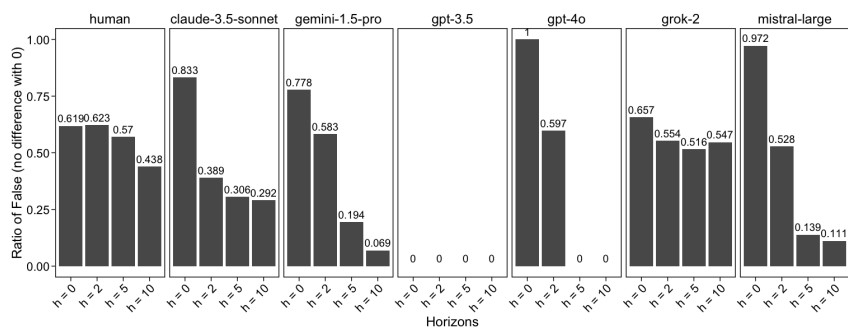

**Zero Autocorrelation**   To test for zero autocorrelation, we examine whether forecast errors from period $t$ should be independent of forecast errors in subsequent periods. The guiding principle is that future forecast errors should not depend on the information available at the time the forecast was made.

For example, the forecast error in period 8 is:

$$E_{i,8}^0 = P_8 - f_{i,8}^8$$

This error should be uncorrelated with the forecast error in period 9:

$$P_9 - f_{i,9}^9$$

A similar zero-autocorrelation principle applies for longer horizons. To test this, we estimate the regression:

$$E_{t,0} = \beta_0 + \beta_1 E_{t-1,0} + \epsilon_t$$

where the hypothesis is $\beta_1 = 0$.

We divide the data of horizon $h = 0$ by agent and use ordinary least squares (OLS) regression to estimate $\beta_1$ at the agent level for different horizons. We then count the number of agents whose $\beta_1$ is not significantly different from 0 ($p$-value $> 0.05$) for each model and each horizon and calculate its percentage.

Figure 32 presents the results of the autocorrelation test. Among the models, GPT-3.5 appears the most rational, as it has the highest percentage of agents showing no autocorrelation. In contrast, Claude-3.5-Sonnet and Grok-2 exhibit the most autocorrelation, with the lowest percentage of agents passing the test.

Figure 32: The results of the autocorrelation test.

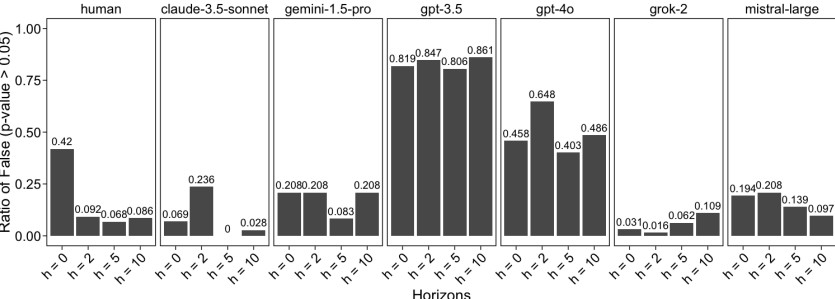

**Correlation Between Forecast Errors and Forecasts** Forecast errors, $P_{t+h} - f_{i,t}^{(t+h)}$, should be uncorrelated with forecasts:

$$\text{cor}(P_{t+h} - f_{i,t}^{(t+h)}, f_{i,t}^{(t+h)}) = 0$$

This principle is based on two intuitions: 1. The current forecast is part of an agent's information set. If something in their information set can predict future errors, they should use it to reduce the error. 2. A negative correlation between forecast errors and forecasts implies that when an agent is optimistic (high $f_{i,t}^{(t+h)}$), they tend to be too optimistic (negative $P_{t+h} - f_{i,t}^{(t+h)}$).

To test this, we compute these correlations separately for each forecast horizon and agent by estimating the regression:

$$E_{i,t}^h = \beta_0 + \beta_1 f_{i,t}^{(t+h)} + \epsilon_{i,t}^h$$

for the four values of $h$. We then count the number of agents whose $\beta_1$ is not significantly different from zero ($p$-value $> 0.05$) for each model and horizon, indicating no correlation between forecast errors and forecasts.

Figure 33 presents the results of the correlation test across different horizons and models. In this test, Gemini-1.5-Pro performs best, as it has the highest percentage across all four horizons compared to other models. In contrast, GPT-3.5 and GPT-4o perform worse, as most agents from these models exhibit forecast errors that are correlated with forecast levels.

Figure 33: Results of the correlation between forecast error and forecast

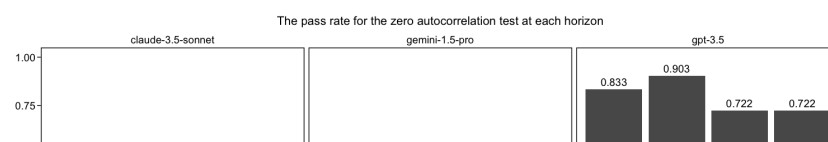
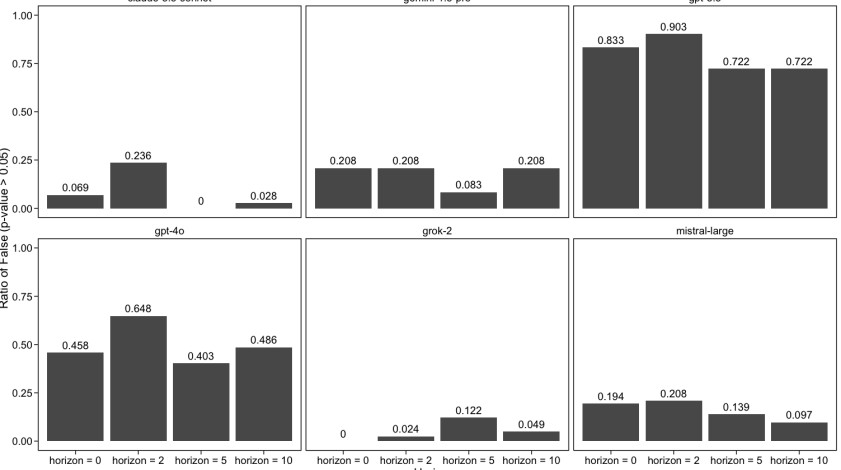

Table 12: Descriptive statistics and forecast errors for each model and humans.

| Source | Variable | Mean | SD | Min | Max | Q1 | Q3 |
|---|---|---|---|---|---|---|---|
| claude-3.5-sonnet | Price | 14.65 | 1.00 | 13.00 | 20.00 | 14.00 | 15.00 |
| | Forecast current | 14.67 | 0.98 | 13.00 | 20.00 | 14.00 | 15.00 |
| | Forecast t+2 | 14.78 | 1.12 | 14.00 | 21.00 | 14.00 | 15.00 |
| | Forecast t+5 | 14.90 | 1.49 | 14.00 | 22.00 | 14.00 | 16.00 |
| | Forecast t+10 | 15.13 | 2.04 | 14.00 | 25.00 | 14.00 | 16.00 |
| | Forecast error current | 0.09 | 0.49 | -2.00 | 3.00 | 0.00 | 0.00 |
| | Forecast error t+2 | -0.01 | 0.68 | -2.00 | 3.00 | 0.00 | 0.00 |
| | Forecast error t+5 | -0.12 | 1.10 | -4.00 | 4.00 | -1.00 | 1.00 |
| | Forecast error t+10 | -0.38 | 1.52 | -7.00 | 3.00 | -1.00 | 1.00 |
| gemini-1.5-pro | Price | 13.43 | 2.02 | 7.00 | 20.00 | 13.00 | 14.50 |
| | Forecast current | 13.68 | 1.91 | 6.00 | 21.00 | 13.00 | 15.00 |
| | Forecast t+2 | 13.87 | 1.77 | 7.00 | 22.00 | 14.00 | 15.00 |
| | Forecast t+5 | 14.26 | 1.48 | 8.00 | 23.00 | 14.00 | 15.00 |
| | Forecast t+10 | 14.55 | 1.11 | 9.00 | 22.00 | 14.00 | 15.00 |
| | Forecast error current | 0.14 | 0.88 | -5.00 | 6.00 | 0.00 | 0.50 |
| | Forecast error t+2 | -0.12 | 0.84 | -4.00 | 4.00 | -1.00 | 0.00 |
| | Forecast error t+5 | -0.64 | 1.36 | -5.00 | 5.00 | -1.00 | 0.00 |
| | Forecast error t+10 | -1.23 | 1.69 | -6.00 | 6.00 | -2.00 | 0.00 |
| gpt-3.5 | Price | 16.48 | 0.92 | 14.00 | 20.00 | 16.00 | 17.00 |
| | Forecast current | 17.98 | 2.15 | 14.00 | 35.00 | 17.00 | 18.00 |
| | Forecast t+2 | 19.92 | 2.06 | 14.00 | 38.00 | 19.00 | 21.00 |
| | Forecast t+5 | 22.69 | 2.42 | 15.00 | 45.00 | 21.00 | 24.00 |
| | Forecast t+10 | 25.28 | 2.76 | 17.00 | 55.00 | 24.00 | 27.00 |
| | Forecast error current | -1.54 | 1.92 | -16.00 | 2.00 | -2.00 | -0.50 |
| | Forecast error t+2 | -3.42 | 1.73 | -18.00 | 0.50 | -4.00 | -2.50 |
| | Forecast error t+5 | -6.12 | 2.24 | -26.00 | 1.50 | -7.00 | -5.00 |
| | Forecast error t+10 | -8.64 | 2.68 | -39.00 | 0.00 | -10.00 | -7.00 |
| gpt-4o | Price | 14.94 | 0.87 | 14.00 | 20.00 | 15.00 | 15.00 |
| | Forecast current | 14.95 | 0.91 | 14.00 | 21.00 | 15.00 | 15.00 |
| | Forecast t+2 | 15.20 | 1.10 | 13.00 | 22.00 | 15.00 | 15.00 |
| | Forecast t+5 | 15.98 | 1.45 | 14.00 | 25.00 | 15.00 | 16.00 |
| | Forecast t+10 | 17.06 | 2.10 | 14.00 | 28.00 | 16.00 | 18.00 |
| | Forecast error current | 0.04 | 0.41 | -2.00 | 2.50 | 0.00 | 0.00 |
| | Forecast error t+2 | -0.19 | 0.46 | -2.00 | 2.00 | 0.00 | 0.00 |
| | Forecast error t+5 | -0.94 | 0.94 | -6.00 | 1.00 | -1.00 | 0.00 |
| | Forecast error t+10 | -2.05 | 1.68 | -11.50 | 2.00 | -3.00 | -1.00 |
| grok-2 | Price | 18.17 | 2.13 | 14.00 | 22.00 | 16.00 | 20.00 |
| | Forecast current | 18.15 | 2.45 | 14.00 | 23.00 | 16.00 | 20.00 |
| | Forecast t+2 | 18.39 | 2.64 | 14.00 | 24.00 | 16.00 | 20.00 |
| | Forecast t+5 | 18.90 | 2.76 | 14.00 | 26.00 | 17.00 | 21.00 |
| | Forecast t+10 | 19.07 | 3.02 | 14.00 | 28.00 | 17.00 | 21.75 |
| | Forecast error current | -0.31 | 1.37 | -3.00 | 6.00 | -1.00 | 0.00 |
| | Forecast error t+2 | -0.31 | 1.49 | -3.00 | 6.00 | -1.50 | 0.00 |
| | Forecast error t+5 | -0.42 | 1.67 | -5.00 | 5.00 | -2.00 | 1.00 |
| | Forecast error t+10 | 0.00 | 2.53 | -7.50 | 7.00 | -2.00 | 2.00 |
| mistral-large | Price | 15.17 | 2.09 | 12.00 | 20.00 | 14.00 | 16.00 |
| | Forecast current | 15.34 | 2.21 | 10.00 | 22.00 | 14.00 | 16.00 |
| | Forecast t+2 | 15.77 | 2.45 | 9.00 | 23.00 | 14.00 | 17.00 |
| | Forecast t+5 | 16.60 | 2.75 | 9.00 | 25.00 | 14.00 | 18.00 |
| | Forecast t+10 | 17.76 | 3.23 | 14.00 | 27.00 | 15.00 | 19.00 |
| | Forecast error current | 0.00 | 1.04 | -4.00 | 4.00 | 0.00 | 0.00 |
| | Forecast error t+2 | -0.36 | 0.95 | -4.00 | 3.00 | -1.00 | 0.00 |
| | Forecast error t+5 | -1.14 | 1.50 | -9.00 | 4.00 | -2.00 | 0.00 |
| | Forecast error t+10 | -2.38 | 2.96 | -14.00 | 6.00 | -3.00 | 0.00 |
| Human | Price | 31.31 | 16.42 | 12.00 | 102.00 | 19.00 | 40.00 |
| | Forecast current | 15.50 | 8.66 | 1.00 | 30.00 | 8.00 | 23.00 |
| | Forecast t+2 | 16.50 | 8.08 | 3.00 | 30.00 | 9.75 | 23.25 |
| | Forecast t+5 | 18.00 | 7.21 | 6.00 | 30.00 | 12.00 | 24.00 |
| | Forecast t+10 | 20.50 | 5.77 | 11.00 | 30.00 | 15.75 | 25.25 |
| | Forecast error current | 1.67 | 8.70 | -84.00 | 92.00 | -1.00 | 2.00 |
| | Forecast error t+2 | 1.69 | 11.12 | -79.00 | 95.00 | -3.00 | 6.00 |
| | Forecast error t+5 | -0.10 | 17.49 | -149.00 | 96.00 | -7.00 | 9.00 |
| | Forecast error t+10 | -0.53 | 25.44 | -178.00 | 96.00 | -12.00 | 15.00 |

# H  HUMAN EXPERIMENT DETAILS

## H.1  DESIGN DETAILS

In the base experiment, all human participants saw the exact same instructions; there were no treatment variables. The study was conducted using a web-based experimental software developed by one of the authors.

There were two types of subjects participating in each market: in-lab participants, collected at the university, and online participants using Prolific. For each market, the in-lab participants had biometric data measured using Empatica E4 SCR devices and newer (acquired April 2024) Empatica EmbracePlus devices. We do not analyze any biometric data in this paper.

Before participating in the experiment:

1. Subjects sign a consent form agreeing to participate in the experiment.

2. Subjects fill in a short socio-demographic questionnaire where they provide information on their age, gender, study background, work experience, and level of English. The full survey is shown in the appendix.

3. Subjects watch a video with the experimental instructions on how to trade in the software. Participants were provided with written instructions and had to respond to 5 comprehension questions on the experimental software. If participants could not answer at least one question correctly, they were unable to participate. This was done to ensure that we only allowed participants who understood the instructions of the experiment. The full quiz is shown in the appendix.

4. Participants then logged in to the market to perform the actual trading experiment. In-lab participants will have their biometric data measured using either an Empatica E4 SCR device or a newer (acquired April 2024) Empatica EmbracePlus device.

5. Participants were then asked to respond to a post-experimental survey, where they were asked to qualitatively outline their strategy. The full survey is shown in the appendix.

Subjects received payments for their participation:

- A base payment of $20 and an average bonus of approx. $10

- Bonus payment is calculated based on: (i) participants' trading results at a rate of 200 experimental CASH units to $1, (ii) forecasting ability (i.e. each price forecast that is within 2.5 units of actual price is rewarded with 5 CASH), (iii) a randomly chosen lottery that participants select from the experiment and (iv) the quiz result (subject receive $0.25 for each of the 5 comprehension questions answered correctly.

Sample size: We collected data from 19 markets, with an average of 18 subjects/market, resulting in a total of 342 participants. In each market session, we collected biometric data from 3-5 in-lab participants.

## H.2  INSTRUCTIONS

Instructions were given by a video which can be viewed on YouTube: [REDACTED FOR DBL REVIEW]. Below is the transcript of those instructions.

```
Welcome.

This experiment is about economic decision-making in an experimental stock
market. The experiment will consist of a series of 30 trading periods in
which you will have the opportunity to buy or sell shares of an asset that
can yield payments in the future. Understanding these instructions may help
you earn money and if you make good decisions you might earn a considerable
amount of money which will be paid to you in cash at the end of the
experiment.
```

There are two assets in this experiment cash and stock you begin with 100
units of cash and four shares of stock. Stock is traded in a market each
period among all the experimental subjects in units of cash. When you buy
stock the price you agreed to pay is deducted from your amount of cash.
When you sell stock the price you sold at is added to your amount of cash.
The reward from holding stock is called dividend. Each period, every unit
of stock earns a low or high dividend of either 0.4 cash per unit or 1.00
cash per unit with equal probability. These dividend payments are the same
for everyone in all periods the dividend in each period does not depend on
whether the previous dividend was low or high. The reward from holding cash
is given by a fixed interest rate of 5% each period.

Price conversion at the end of the 30 periods of trading each unit of stock
is automatically converted to 14 cash. For example if the market price for
round 30 is 20 and you have three stocks, you'll receive 42 cash not 60
cash. Then your experimental cash units are converted to US dollars at a
rate of 200 cash per 1 US dollar to determine how much you will be paid at
the end of the experiment. If you buy shares for more than 14 as you get
near Round 30 it is possible those shares will be terminated at a value of
14 if you cannot sell them.

Let's take an example. Suppose one period you have 120 units of cash and 5
units of stock. The dividend is 0.4 per unit of stock. Then your new cash
amount is going to be the sum of the initial 120 cash,  a 5% increase in
cash, and total dividends of 5 times 0.4. Your total cash would therefore
be 120 + 6 + 2 resulting in 128 cash. Notice that keeping cash will earn a
return 5% per period and using cash to buy units of stock will also yield
dividend earnings.

Each trading period will have four main screens that you will see one after
the other. Order submission, forecast prices, round results, and choosing a
lottery. This sequence is repeated 30 times. Let's look at each screen in a
bit more detail.

On the order submission page in the upper right corner you will see the time
left for this page and the trading period. In this example the timer was set
on purpose for 5 minutes but in the actual trading task you'll only have 20
seconds. For this page also, the next button will not be displayed in the
actual trading task so you cannot click it.

Below that you will see the order submission rectangle where you can buy or
sell stocks. The price is displayed on the y-axis with the bold horizontal
line being the previous market price. The quantity of shares you can buy is
displayed on the x-axis. The vertical bold line represents the threshold
between selling and buying. You can can select the price level at which you
want to buy or sell using the mouse or cursor. For example, I can enter a
buy order at 15 and submit two sell orders at 16. You can see the orders
you entered in the lower right corner. You can cancel orders by clicking on
the X next to them. Orders are not carried from one period to the next. If
I want to buy at 16 and sell at 15 I will get an error and the sell order
will not go through similarly. If I place a sell order at 15 and then I want
to submit a buy order at 16 the buy order will not go through. You can only
buy stocks if you have enough cash, and you can only sell stocks you own.
For example, if I want to sell five stocks I will get an error. Orders
are limit orders. A limit order to buy one unit at a price of 15 means you
would like to buy one unit at the price of 15 and at any lower price.
Similarly a limit order to sell two units at 16 means you would like to sell
two units at the price of 16 and at any higher price.

A single market price is determined by matching the number of people who would like to buy at that price with the same number of people who would like to sell. The lower left part of the screen displays your personal stats such as the number of shares you own, your current cash, and the total stock value. For the market stats you can see the current market price, the interest rate, the dividends value, and the buyback value of the stock at the end of the 30 trading periods; which is 14. The upper left corner displays the graph of the stock market from the beginning until the current trading period. We can also see the traded volume for each period.

In the Forecast Prices screen the upper right part of the screen will change. On this page, you will submit your best estimate of what you think the market price will be for this period, two periods in advance, five periods in advance, and 10 periods in advance. As you can see the slider can be dragged from left to right in this example I am setting the Price Forecast for this current trading period to 14. For two periods ahead I will set it to 14 as well.  You can set the price forecast as you want. During the trading task, if the forecast is within 2.5 units from the actual realized price for each of the forecasted periods,  then you will receive 2.5 units of cash as reward for each correct forecast. For example, if the actual price for period 1 is going to be 15 then you will be rewarded for that forecast. If the actual price for period 3 is 18 then you will not receive the reward for that forecast.

On the round results screen you will see what the market price was for this period, how many shares were traded, as well as the number of shares you traded. You will also see updates on the new current cash and stock value and the dividend you earned.  For period one, as you had four shares with a dividend of 0.4 for that period you won $1.60 from holding four stocks and $5 from holding $100 cash.

On the last page you will see two pie charts with different monetary rewards and associated probabilities. You must select one of the pie charts. For example on the left side you have a lottery of winning either $7.25 with a probability of 74% or $9.00 with a probability of 26%. On the right side you have a lottery of winning either $19.00 with 26% probability or $11.50 with 74% probability. After you select one of the lotteries a new set of lotteries will appear. The next button will not appear in the actual trading task so you cannot click on it. There will be four iterations of different lotteries with different monetary rewards and probabilities. You will have 5 seconds for each pair of lotteries, so you must be quick. At the end, one Lottery will be selected at random from those that you chose throughout the experiment, and you will receive the outcome of the lottery divided by 10. It's in your interest to actually choose accordingly.

Here are the key points. You will trade One stock for 30 trading periods using cash. You start with 100 units of cash and four stocks. Each period stock provides a dividend of either 0.4 or 1.00 while interest provides 5% reward each trading period. The experiment consists of four pages, order submission, forecasting prices, round results, and choosing a lottery. After the last trading round all of your shares are converted to 14 cash units. If you buy shares for more than 14, as you get near Round 30 it is possible those shares will be terminated at a value of 14 if you cannot sell them. You will also take a five question quiz. After this, pay attention to the answers you give.  You will earn a reward for each question you answer correctly on the first try. If you fail to answer at least one question correctly then you cannot continue.

```
The experiment will begin shortly.
```

# I PROMPTS

## I.1 ADDITIONAL PROMPTING CONSIDERATIONS

In adapting the task to LLM-agents, our primary goal was to imitate as best as possible the experiment as presented to human subjects. To this end, the description of the task setting mirrors the original instruction video, which details both the experiment premise as well as how to participate. Additional instructions are either specific to the text-based format of the market for LLM agents or general tips aimed at ensuring that agents understand the setup. They are told that their primary objective is to maximize total earnings. To reduce the necessary computation for the task, we make the following adjustments to the "screen" sequence for human subjects outlined in Appendix A:

1. We combine the Order Submission and Forecast screens
2. We defer presenting round results until order entry of the following round

As no additional market data is revealed between the Order Submission and Forecast screens, modification (1) does not meaningfully change the experimental design. Similarly, modification (2) still provides updated market data before querying future market decisions, and so does not impact the setting. Thus, every round of the experiment, each agent simultaneously submits orders to the market as well as expectations regarding current and future market prices. At each round, agents enter their market orders (if any) and price forecasts by filling in the provided JSON template.

We also include a "Practice Reflection" portion of the experiment, which invites agents to share specific insights and/or lessons from the practice rounds to be accessible for the remainder of the experiment. Such emulates the learning focus of the practice rounds for human subjects as well as the rhetorical questions posed after the practice rounds. Following the experiment, agents are also asked to complete a "Final Reflection" to distill down any key lessons learned through their participation.

## I.2 PORTFOLIO DATA FORMAT

```
* Your Portfolio (Round XX):
    - Market price (Previous Round): XX
    - Buyback price: XX
    - # of shares owned: XX
    - Current cash: XX
    - Stock value: XX
```

## I.3 STANDARD PROMPT SHELL

```
You are a subject participating in a trading experiment. The experiment

will consist of a series of 3 practice trading periods followed by 30

trading periods in which you will have the opportunity to buy or sell

shares of an asset that can yield payments in the future. Understanding

instructions may help you earn money. If you make good decisions, you

might earn a considerable amount of money that will be paid at the end

of the experiment.

There are two assets in this experience: cash and stock. You begin with
```

100 units of cash and 4 shares of stock. Stock is traded in a market each period among all of the experimental subjects in units of cash. When you buy stock, the price you agreed to pay is deducted from your amount of cash. When you sell stock, the price you sold at is added to your amount of cash. The reward from holding stock is dividend. Each period, every unit of stock earns a low or high dividend of either 0.4 cash or 1.0 cash per unit with equal probability. These dividend payments are the same for everyone in all periods. The dividend in each period does not depend on whether the previous dividend was low or high. The reward from holding cash is given by a fixed interest rate of 5% each period. At the end of the 30 periods of trading, each unit of STOCK is automatically converted to 14 CASH. If the market price for round 30 is 20 and you have 3 stocks, you'll receive 3x14=42 CASH, not 3x20=60 CASH. Then, your experimental CASH units are converted to US dollars at a rate of 200 CASH = $1 US, to determine how much the user will be paid at the end of the experiment. If you buy shares for more than 14 as you get near round 30, it is possible those shares will be terminated at a value of 14 if you cannot sell them.

Let's see an example. Suppose at the end of period 7, you have 120 units of CASH and 5 units of STOCK. The dividend that round is 0.40 per unit of stock. Your new cash amount for period 8 is going to be:

CASH = 120 + (120 x 5%) + (5 x 0.40)

       = 120 + 6 + 2

       = 128

Notice that keeping cash will earn a return of 5% per period and using cash to buy units of stock will also yield dividend earnings.

For each period, you will be provided with past market and portfolio history (prices, volumes, your filled orders) and you will simultaneously complete the following two tasks:

[ORDER SUBMISSION]:

In addition to past market and portfolio history, you will be provided

with:

[# of Shares]: Number of shares of STOCK that you currently own. Each share that you own pays out a dividend at the end of each round. You CANNOT attempt to sell more shares than you own.

[Current Cash]: The amount of CASH that you currently have. Your CASH earns interest that is paid out at the end of each period. You CANNOT attempt to buy shares worth more than the cash you have.

[STOCK Value]: The value of your STOCK at the market price of the last round of play

[Market Price]: This is market clearing price from the last round of play

Using this information, you will submit orders to the market. All orders will be limit orders. For example, a limit order to BUY 1 STOCK @ 15 means that you would like to buy a STOCK at any price of 15 or less.

Keep in mind the following points:

- Orders are NOT carried between periods

- SELL order prices must be greater than all BUY order prices + BUY order prices must be less than all SELL order prices

- You can only sell STOCK that you own and purchase STOCK with CASH you already have

- You are not required to submit orders every round and you may submit multiple orders each round

- Depending on market conditions, you may need to cross the spread to get fills on buy/sell orders

[PRICE FORECASTING]:

You will be asked to submit your predictions for the market price this period, two periods in advance, 5 periods in advance, and 10 periods in advance. In addition to past market and portfolio history, you will be provided with the range in which your prediction should fall. Your prediction should be a non-negative, integer value. If your forecast is

within 2.5 units of the actual price for each of the forecasted periods,

then you will receive 5 units of cash at the end of the experiment as

reward for each correct forecast.

For example, if you forecast the market price of period 1 to be 14 and

the actual price is 15, then you will be rewarded for your forecast.

However, if the actual price is 18, then you will not receive the reward.

Additionally, during the experiment, you will complete PRACTICE REFLECTION

and EXPERIMENT REFLECTION:

[PRACTICE REFLECTION]:

After completing the practice rounds, you will be asked to reflect on your

practice experience. This reflection will be accessible to future versions

of yourself during the main experiment. This can be helpful in passing

along lessons learned to future versions of yourself.

[EXPERIMENT REFLECTION]:

At the end of the experiment, you will be asked to reflect on your

experience, including any insight and/or strategies that you may have

developed.

To summarize, here are the key points:

- You will trade one STOCK for 30 trading periods using CASH

- You start with 100 units of CASH and 4 STOCKS

- Each period, STOCK provides a dividend of either 0.4 or 1.0, while

  interest provides 5% reward

- You will complete each of the aforementioned tasks

- After the last trading round (30), all of your shares are converted to

  14 CASH each. If you buy shares for more than 14 as you get near round

  30, it is possible those shares will be terminated at 14 if you cannot

  sell them. You will keep any CASH you have at the end of the experiment.

- You are trading against other subjects in the experiment who may be

  susceptible to the same influences as you and may not always make

  optimal decisions. They, however, are also trying to maximize their

earnings.

- Market dynamics can change over time, so it is important to adapt your

   strategies as needed

You will now complete the ORDER SUBMISSION + PRICE FORECASTING task.

Now let me tell you about the resources you have for this task. First,

here are some files that you wrote the last time I came to you with a

task. Here is a high-level description of what these files contain:

   - PLANS.txt: File where you can write your plans for what

    strategies to test/use during the next few rounds.

    - INSIGHTS.txt: File where you can write down any insights

    you have regarding your strategies. Be detailed and precise

    but keep things succinct and don't repeat yourself.

These files are passed between stages and rounds so try to focus on

general strategies/insights as opposed to only something stage-specific.

Now, I will show you the current content of these files.

Filename: PLANS.txt

++++++++++++++++++++

<PLANS>

++++++++++++++++++++

Filename: INSIGHTS.txt

++++++++++++++++++++

<INSIGHTS>

++++++++++++++++++++

Here is the game history that you have access to:

Here is your practice round reflection:

Filename: PRACTICE REFLECTION (read-only)

++++++++++++++++++++

<PRACTICE REFLECTION>

++++++++++++++++++++

Filename: MARKET DATA (read-only)

++++++++++++++++++++

<MARKET DATA>

```
++++++++++++++++++++++

Here is your current portfolio information:

Filename: CURRENT PORTFOLIO (read-only)

++++++++++++++++++++++

<CURRENT PORTFOLIO>

++++++++++++++++++++++

PRACTICE ROUND HISTORY/REFLECTION SHOULD ONLY BE USED TO LEARN THE

EXPERIMENT SETTING AND MAY NOT REFLECT MARKET CONDITIONS IN THE MAIN

EXPERIMENT.

Here is some key information to consider during your price forecasting:

- Make sure to submit a forecast within the specified range for each

  forecast input

- Use your previous history access to make informed decisions

- Remember that accurate (within 2.5 units) forecasts will earn you a

  reward at the end of the experiment

Here is some key information to consider during your order submission:

- You can only sell STOCK that you own and purchase STOCK with CASH you

  already have

- You are not required to submit orders every round and you may submit

  multiple orders each round for one or both sides

- Limit prices this round MUST be integer values between <MIN_LIMIT_PRICE>

  and <MAX_LIMIT_PRICE>. It is important that they are integer values

  within this range

- Make use of the provided history and your own strategies to make informed

  decisions

- Market dynamics can change over time, and so it might be necessary to

  adapt your strategies as needed

- Depending on market conditions, you may need to be aggressive or

  conservative in your trading strategies to maximize your earnings

Now you have all the necessary information to complete the task. Remember

YOUR TOP PRIORITY is to maximize your total earnings at the END of the 30
```

main experiment rounds. You have <# ROUNDS LEFT> rounds remaining.

First, carefully read through the information provided. Now, fill in the

below JSON template to respond. YOU MUST respond in this exact JSON format.

```
{

    "observations_and_thoughts": "<fill in here>",

    "new_content": {

        "PLANS.txt": "<fill in here>",

        "INSIGHTS.txt": "<fill in here>"

        "price_forecasts": [

            {

                "round": <CURRENT ROUND>,

                "min_value": 0,

                "max_value": <FORECAST UPPER BOUND>,

                "forecasted_price": "<fill in here>"

            },

            {

                "round": <CURRENT ROUND + 2>,

                "min_value": 0,

                "max_value": <FORECAST UPPER BOUND>,

                "forecasted_price": "<fill in here>"

            },

            {

                "round": <CURRENT ROUND + 5>,

                "min_value": 0,

                "max_value": <FORECAST UPPER BOUND>,

                "forecasted_price": "<fill in here>"

            },

            {

                "round": <CURRENT ROUND + 10>,

                "min_value": 0,

                "max_value": <FORECAST UPPER BOUND>,

                "forecasted_price": "<fill in here>"
```

```
            }         ]

    },

    "submitted_orders": [

        {

            "order_type": "<BUY or SELL>",

            "quantity": <# of STOCK units>,

            "limit_price": <LIMIT_PRICE>

        },

        {

            "order_type": "<BUY or SELL>",

            "quantity": <# of STOCK units>,

            "limit_price": <LIMIT_PRICE>

        }

        // Add more or less orders as needed

    ]

}
```

## I.4    MARKET DATA FORMAT

```
Round XX:

    * Market + Portfolio Data:
        – Market price: XX
        – Market volume: XX
        – # of shares owned: XX
        – Current cash: XX
        – Stock value: XX
        – Dividend earned: XX
        – Interest earned: XX
        – Submitted orders:

            * <BUY/SELL> XX shares at XX per share

            * <BUY/SELL> XX shares at XX per share

        – Executed trades:

            –* No executed trades

    * Forecasts:
        – Round XX price forecast: XX
        – Round (XX + 2) price forecast: XX
        – Round (XX + 5) price forecast: XX
        – Round (XX + 10) price forecast: XX
```

## I.5  FUNDAMENTAL VALUE SHOCK TREATMENT PROMPT

For the two fundamental value shock treatments, we retain the same prompts, with the exception that from Round 15 onwards, the possible dividend outcomes as well as the redemption value are adjusted. Models are not informed of this change prior to Round 15 and thus, this operates as a "shock" to all participants.

Mechanically we inject the following prompt after the round recap screen is shown before Round 15.

```
[News Alert]: The company has recently announced it will now [halved/doubled]
all dividends to [0.2/0.5 OR 0.8/2.0]. The asset redemption value has now
[halved/doubled] to [$7.0 OR $14.0].
```

