# OpenReview forum: "LLM Agents Do Not Replicate Human Market Traders: Evidence from Experimental Finance"
_ICLR.cc/2026/Conference — Submitted to ICLR 2026_

### Official Review · Reviewer_KABm · 2025-10-21

**Soundness:** 2
**Presentation:** 2
**Contribution:** 2
**Rating:** 2
**Confidence:** 4

**Summary:**

This study compares Large Language Models (LLMs) with human traders in experimental markets where prices emerge endogenously. LLMs consistently price assets near their fundamental values and show little bubble formation, while humans generate substantial bubbles. These patterns persist across different market conditions. Analysis of LLM strategies shows lower variance, reduced bias, and stronger reliance on fundamentals. The authors conclude that LLMs are poor proxies for modeling human market behavior, as they do not replicate key phenomena like large bubbles.

**Strengths:**

1. The findings show clear behavioral differences: LLM-driven markets exhibit far more “textbook‑rational” pricing than human‑driven markets.
2. The paper shows that LLMs incorporate less bias into their price forecasts and rely more on fundamental‑value strategies, unlike the heuristic‑driven approaches common among human traders.
3. These results challenge the assumption that “out‑of‑the‑box” LLMs can reliably replicate human market dynamics, especially the emergence of phenomena like bubbles and crashes.

**Weaknesses:**

1. The experimental setup appears overly simplified, which undermines the strength of the authors’ claims. The simulated market environment does not adequately capture the complexity of real-world trading scenarios, making it difficult to draw robust conclusions about performance differences between human traders and LLMs.
2. The configuration of LLM agents in the experiment also seems simplistic and may not fully reflect the models’ ability to understand and develop trading strategies. The implementation approach resembles prompt engineering, but it is unclear to what extent prompt design was influenced by human intervention or biases, which could affect the validity of the results.
3. The paper does not provide sufficient detail about the human participants involved in the study. Without clear information on their backgrounds, trading experience, or demographic characteristics, it is difficult to assess the credibility of the reported differences and the generalizability of the findings.

**Questions:**

See the weaknesses.

---

> ### Author Response · Authors · 2025-11-20
>
> Thank you for the careful review and for highlighting the strengths of our empirical comparisons. We address the concerns raised below.
>
> 1. “The experimental setup is overly simplified.”
>
> The simplicity is intentional and aligns with the canonical methodology of experimental finance.
>
> The Smith–Suchanek–Williams market is specifically designed to isolate endogenous deviations from value and has been the foundation for 40 years of bubble research. Nearly all major findings in experimental finance, bubbles, crashes, momentum heuristics, overconfidence, and herding, were discovered in similarly simple, tightly controlled environments.
>
> There is an established precedent for ICLR behavioral-LLM papers (Binz ’24; Rozen ’25; Liu ’25), which likewise begin with stylized benchmarks before adding complexity. Our work fits within this line of research, all of which begin with stylized canonical tasks before layering in complexity. These papers argue that establishing clean baselines is essential before interpreting LLM behavior in richer environments.
>
> Our logic is identical:
>
> If LLMs fail to reproduce human behavior in the simplest and most diagnostic environment, where human bubbles are largest and most robust, then it is unlikely that human-like behavior will emerge in more complex markets.
>
> This makes the stylized design a feature, not a limitation.
>
> To test robustness, we also included two small perturbations (mid-experiment dividend shock; repeated-session runs), and the qualitative result persisted.
>
> 2. “LLM configuration seems simplistic / prompt-engineering issues.”
>
> LLMs receive the same information and incentives as humans: the dividend process, the interest rate, the redemption rule, and the payoff structure. We do not provide FV explicitly.
>
> This exactly mirrors human instructions in the experimental-finance literature.
>
> Our design is intentionally minimal for the same reason classic experimental-finance markets are minimal:
> To abstract away institutional frictions and break decision-making down to its simplest components, so we can see what is happening “under the hood.”
>
> This abstraction is what allows the field to diagnose core behavioral mechanisms like speculation, overconfidence, or momentum trading without confounds from complex market microstructure.
>
> During development, we tested a range of modest prompt variations—persona prompts (“retail trader”), momentum/heuristic framings, emotional priming (FOMO), and even variations proposed by the models themselves. These altered the reasoning style, but did not generate human-scale bubbles unless the model was explicitly instructed to “create a bubble.” We intend to do a more systematic review of different prompting techniques in future work.
>
> This strongly suggests the result is not a prompt artifact, but rather reflects a deeper behavioral tendency of LLM agents in endogenous-price environments.
>
> 3. “Insufficient detail on human participants.”
>
> Our human dataset consists of a mixed pool of university subjects and Prolific participants, a standard combination for modern experimental economics. We have detailed demographic and trading-experience data, which we can add to the camera-ready appendix.
>
> It is important to note that professional traders also generate large bubbles in this market (Weitzel et al., 2019), indicating that trading experience and sophistication alone are insufficient to prevent bubble formation. The divergence observed with LLMs is therefore meaningful rather than a product of our subject pool.
>
> 4. Summary and Contribution
>
> Our contribution is not simply that “LLMs behave rationally.”
> It is that LLM agents fail to reproduce the most robust human phenomenon in the benchmark experimental-finance environment: large speculative bubbles.
>
> This divergence appears despite:
>
> -Identical information and incentives,
>
> -An environment explicitly designed to elicit bubbles,
>
> -Competing behavioral templates available in model training data (textbook valuation, human speculation, rational-trend strategies)
>
> - Multiple model families (GPT, Claude, Gemini, Mistral, Grok), prompt variations, and initial perturbations.
>
> This gives us the first clean baseline for how LLMs behave when prices are fully endogenous. Establishing such baselines is standard in behavioral-LLM research at ICLR and necessary before introducing complexity such as uncertainty, multi-asset markets, communication, and mixed human–LLM settings.
>
> Our work therefore fits directly within the existing line of behavioral-LLM research at ICLR—methodologically, in motivation, and in contribution.

---

### Official Review · Reviewer_tK82 · 2025-10-27

**Soundness:** 2
**Presentation:** 2
**Contribution:** 1
**Rating:** 2
**Confidence:** 4

**Summary:**

The paper conducted LLM experiments on experimental finance settings where there's a known fundamental value and compare it with human experimental data. The paper finds that LLM stick close to fundamentals and rarely form large bubbles, which departs from human behaviors. The paper also tested settings on battle royale markes and forecasts, showing LLM-human differences.

**Strengths:**

The paper covers several topics, there's comparison to human baselines and careful analysis.

**Weaknesses:**

I'm a bit unsold on the motivation and contributions.

- First of all, the exact types of experimental data and 'correct'/'expected' behavior is well within the training data of the LLM. In addition, prompts in the experiment disclose redemption value / fundamental value mechanisms, so it is largely unsurprising to observe the discovered LLM phenomenon. In other words, "textbook-rational" seem unsurprising given that LLMs are trained on the said textbooks, as well as the CoT reasoning paradigm that they are trained to use.
- Given the above, I’m not sure that the mere discovery of LLM "rationality" is new or persuasive for this community. Nowadays, models already hit near–top human accuracy on competitive math (gold medal in IMO) and similar benchmarks; so the fact that an LLM looks more rational than the average human-especially when the task is framed explicitly with a redemption value-doesn’t feel surprising. I do see value in documenting this within experimental finance, but the acceptance bar likely requires more than this.
- Most experiments are run on (or include) older model generations. In line 1028 the authors noted that "Grok 2 has been deprecated, and
thus we were unable to run this treatment with that model. Gemini-1.5 had technical errors that will be rerun for camera-ready submission". I think especially given the framing of the papers in line 450, any deployment in real markets will minimally use one of the up to date reasoning models.

Some minor points:
- Humans face strict per-screen time limits (20 seconds to trade and 30 seconds to forecast). There seem to be a difficulty translating that to time and budget for LLMs. In the prompt used in the paper, the authors described (line 971) "PLANS/INSIGHTS" files that agents make on top of chain of thought. That's a notable difference in experimental conditions.
- There are some inconsistent findings across models (e.g., Gemini-1.5 Pro vs. GPT-3.5 vs. GPT-4o), suggesting model-specific idiosyncrasies that may not hold across generations. The paper’s own summary tables/plots show materially different error patterns and dynamics by model.
- The paper also notes "Token usage for Gemini 1.5 Pro could not be accessed through the API console". This is a minor point but is not consistent with my experience with using it.

**Questions:**

- LLMs have inherent prompt sensitivity. Is there any ablations on how that will affect the results (e.g. one counterargument from the papers that 'claim' to use LLM for human simulation might say that they are explicitly asking LLM to take on human persona and roles - will that change behaviors systematically)

- Can the authors articulate the contribution beyond documenting "LLMs look rational under explicit FV prompts"? What is the core conceptual/empirical advance for experimental finance or agent-based markets, and how does it differ from prior work and adds new insights to the community?

- Results may hinge on PLANS/INSIGHTS + CoT and looser time budgets for models. Is there results on no-memory/no-cot agents, or, another extreme, on more up-to-date advanced reasoning models?

---

> ### Author Response · Authors · 2025-11-20
>
> Thank you for your detailed review and for highlighting the strengths of the empirical comparison. We address the points raised below.
>
> 1. Contribution Beyond “LLMs Look Rational”
>
> Our contribution is not that LLMs behave rationally.
> It is that LLM agents fail to reproduce the central human phenomenon in canonical experimental finance: large speculative bubbles.
>
> Information parity:
> We do not provide FV explicitly; we give the same ingredients that humans receive. In the Smith–Suchanek–Williams market, humans reliably form bubbles despite having the exact information needed to compute FV. Thus, FV-anchoring is not a prompt artifact but a genuine behavioral divergence.
>
> Why this was not predetermined.
> LLMs’ training corpora contain conflicting behavioral templates:
> -textbook valuation,
> -human bubble dynamics,
> -rational destabilizing strategies (De Long et al., 1990).
> It was not obvious ex ante which mode would dominate. The fact that models consistently select the pattern humans almost never exhibit (strict FV anchoring) is, in itself, a conceptual contribution.
>
> Additional contributions:
> -Evidence LLMs do not substitute for humans in a domain with well-mapped human behavior.
> -First controlled multi-agent LLM market with endogenous prices.
> -Direct, like-for-like comparison of LLMs and humans performing identical tasks under identical rules—rare in prior work and crucial for interpreting behavioral gaps.
> -A clean baseline for future work (uncertainty in FV, multi-asset markets, communication, mixed human–LLM markets, prompting interventions, fine-tuning).
>
> There is an established precedent for ICLR behavioral-LLM papers (Binz ’24; Rozen ’25; Liu ’25), which likewise begin with stylized benchmarks before adding complexity. Our work fits within this line of research.
>
> 2. Why Start With the Stylized Market (Simplicity Is the Feature)
>
> The stylized market is the canonical diagnostic environment because it isolates endogenous deviations from value. This mirrors the methodology used by behavioral-LLM papers at ICLR, which start from canonical tasks before exploring richer settings.
>
> Our logic is the same:
>
> If LLMs do not reproduce human behavior in the simplest environment, where bubbles are clearest, it is unlikely they will do so in more complex settings.
>
> We also ran two perturbations (dividend shock; experienced-agent repeats), and the qualitative result persists.
>
> 3. Model Recency, Differences, and Why a Baseline Matters
>
> We agree recency matters. In our data, the newest models (GPT-4o, Claude-Sonnet, Gemini-1.5-Pro) show the strongest FV anchoring; older models show more noise. Thus, including older models weakens the main result rather than creating it.
>
> Cross-model variation is informative: if future models begin deviating from FV or becoming more human-like, that would be scientifically notable. A clean baseline now is what makes tracking such shifts possible.
>
> 4. PLANS/INSIGHTS, CoT, and Time Budgets
>
> Humans rarely use their full screen-time windows (avg input of 11.7s; we will add a histogram to the camera-ready appendix); LLMs produce output far below token limits. PLANS/INSIGHTS offer lightweight memory scaffolding common in LLM-agent work, though we agree some asymmetry exists.
>
> The main qualitative effect, FV anchoring and the absence of bubbles, remains stable across all models and prompt variants we report. While we did not include no-memory or strict-limit ablations, we do not believe the result hinges on these scaffolds. We are open to adding variants in the camera-ready if helpful.
>
> 5. Prompt Sensitivity
>
> We explored several prompt perturbations (persona, heuristic/momentum, emotional/FOMO, and prompts proposed by the models themselves). These changed explanations but not trading behavior: models continued to track FV and did not generate human-scale bubbles unless explicitly instructed to do so.
>
> These informal checks suggest the core result is not prompt-sensitive; a systematic study is a natural extension.
>
> 6. Cross-Model Consistency
>
> While quantitative differences exist, the same qualitative pattern holds across all models:
>
> -strong FV anchoring,
>
> -absence of human-scale bubbles,
>
> -strategy language, unlike humans.
>
> These shared regularities are what matter for the conceptual point.
>
> Summary
>
> Our main contribution is not that “LLMs act rationally.”
> It is that LLM agents fail to reproduce the single most robust human pattern in the benchmark experimental-finance environment: speculative bubbles, despite identical information, incentives, standard elicitation conditions, competing behavioral templates, and multiple model families.
>
> This establishes the first clean baseline for understanding how LLM agents behave when prices are fully endogenous, essential for future work on richer market settings and for interpreting future model generations.
>
> The approach mirrors established behavioral-LLM work at ICLR and fits squarely within the conference’s interests.

---

> > ### Comment · Reviewer_tK82 · 2025-11-22
> >
> > Thank you for the response. I have read the rebuttal, but I remain unconvinced regarding the core contribution and impact of the work, so I want to follow up with my main concern, which remains the same as in my original review.
> >
> > - You stated in the rebuttal:
> > >Our contribution is not that LLMs behave rationally. It is that LLM agents fail to reproduce the central human phenomenon in canonical experimental finance: large speculative bubbles.
> >
> > I find this distinction difficult to follow. In the Abstract, you explicitly state that your findings reveal LLMs exhibit a "textbook-rational" approach. In the context of this specific experimental designn- where the "rational" behavior is strictly defined as tracking the Fundamental Value - the absence of bubbles is a direct, deterministic consequence of that rationality. If an agent is "textbook-rational," they will by definition not generate speculative bubbles. Therefore, these appear to be the same finding, which brings me back to my original concern: if LLMs are trained on textbooks that define this rationality, this result feels expected.
> >
> > -  Your argument for significance relies on the premise that the community currently expects LLMs to function as accurate human simulators in economic contexts. I am skeptical that this is the mainstream consensus in the CS/ML community. While LLMs are powerful, they are widely understood to be trained on high-quality data (including textbooks and rational problem-solving), and they excel at tasks like math competitions (e.g. IMO) that are far beyond average human capability. The paper cites works like Horton (2023) and Leng (2024) to suggest a trend of using LLMs as substitutes, but these appear to be Economics/Social Science working papers, with the first paper in particular published in Jan 2023, well before the recent developments of LLMs and deep reasoning models. In addition, Leng (2024) already notes in their abstract that "However, the models also diverge from human subjects in notable ways—most prominently, they maintain a conservatively cautious stance in house‐money or break‐even gambles, rarely opting for higher‐risk choices.", which does not seem to be arguing that LLMs are accurate human behavior proxy. To demonstrate that this paper solves a problem and/or corrects a wrong perspective/assumption relevant to the ICLR community, it will be helpful for the authors to provide:
> >     - Concrete evidence from ML venues (published in main proceedings of ICLR/NeurIPS/ICML) where the main assumption is that base-model LLMs (without specific persona fine-tuning) should naturally replicate human behavioral irrationalities like market bubbles or similar.
> >     - Full citations for the papers mentioned in your rebuttal: "Binz ’24; Rozen ’25; Liu ’25." These do not appear to be listed in the paper's bibliography, and short citations are insufficient for me to verify the claim that there is an established precedent for this specific type of behavioral work at ICLR.
> >
> > Without evidence that the ML community actually holds this "simulator" assumption, the finding that LLMs revert to their training data (rationality) rather than simulating human error remains, in my view, a limited and not surprising contribution.

---

### Official Review · Reviewer_Jkb4 · 2025-10-28

**Soundness:** 3
**Presentation:** 3
**Contribution:** 3
**Rating:** 8
**Confidence:** 4

**Summary:**

Experimental Design: Conducts homogeneous (single-model) and heterogeneous ("Battle Royale") LLM-agent markets.

Behavioral Divergence: Demonstrates that LLMs exhibit "textbook-rational" behavior, while humans consistently generate price bubbles.

Robustness Tests: Validates findings across dividend shocks, experienced sessions, and linguistic analysis of trading strategies.

Forecast Rationality: Shows LLMs produce more accurate and less biased price forecasts than humans.

**Strengths:**

1.  First systematic comparison of LLMs and humans in endogenous experimental markets, bridging behavioral finance and AI alignment

2. Rigorous methodology: a lot of markets vs. 6 LLM models (Claude-3.5, GPT-4o, etc.), with controls for dividend shocks and experience.

3. Well-structured with clear visualizations and statistical tests.

4. Challenges the use of off-the-shelf LLMs as human proxies in finance experiments.

**Weaknesses:**

Are larger models (e.g., GPT-4 Turbo, Claude 3 Opus) and LLMs that may exhibit different behaviors excluded?

Why LLMs are anchored to fundamentals is not explored.

The simplified single-asset design lacks real-world characteristics (e.g., short selling, information asymmetry).

The incentives of human participants (e.g., monetary rewards) may not align with the "profit maximization" imperative of LLMs.

The impact of imperative engineering (e.g., explicit bubble-inducing directives) is not explored.

**Questions:**

Would a larger LLM (e.g., Claude 3 Opus) or a modified RLHF model exhibit bubble-like behavior?

How do the results apply to markets involving high-frequency trading or multi-asset portfolios?

Does the human participants' prior financial knowledge influence the results?

Can LLMs more accurately simulate inexperienced traders?

---

> ### Author Response · Authors · 2025-11-20
>
> Thank you for the thoughtful and positive review. We appreciate the strong assessment of the paper’s contribution and address the raised concerns below.
>
> 1. Use of Larger or Newer Models
>
> We agree that evaluating frontier models (e.g., Claude 3 Opus, GPT-4 Turbo) is valuable. Our aim in this paper was to establish a broad cross-section of model families (GPT, Claude, Gemini, Mistral) to identify architectural regularities and divergences. This breadth helps ensure generalizability.
>
> Importantly, the pattern we document does not rely on weaker models: newer models (GPT-4o, Claude-Sonnet, Gemini-1.5-Pro) show the strongest FV anchoring, while older models deviate more and occasionally display small bubbles.
>
> Thus, including older models attenuates the main result rather than producing it.
>
> We will clarify this framing in the revision and note that future models diverging from fundamentals would itself be a scientifically important finding. This paper provides the necessary baseline for tracking such shifts.
>
> 2. Why LLMs Are Anchored to Fundamentals
>
> Understanding why LLMs anchor strongly to FV is an important dimension, and we plan to expand this discussion.
>
> Based on our textual and behavioral analyses, several hypotheses emerge:
>
> -Inductive bias: Reasoning-oriented models default toward analytical, FV-based explanations.
>
> -Training mixture: LLM corpora include heavy exposure to “textbook” valuation logic, alongside, but not dominated by, descriptions of human speculative bubbles.
>
> -Internal consistency: LLM forecasts and trades are mutually reinforcing in a way that suppresses volatility.
>
> We will make these possible mechanisms more explicit, emphasizing that the paper’s main contribution is documenting this behavioral signature in a controlled environment where human behavior is well understood.
>
> 3. Simplified Market Design
>
> The stylized single-asset design is standard in experimental finance (Smith–Suchanek–Williams, 1988) because it isolates endogenous mispricing and produces the clearest and most measurable bubble dynamics.
>
> Our goal in this paper is to:
>
> -Establish the foundational baseline for LLM behavior in the simplest diagnostic environment. If LLMs do not reproduce human bubbles under these conditions, it is unlikely they will do so in more complex settings.
>
> It is also worth emphasizing that many of the most influential results in behavioral finance, momentum trading, mispricing, experience effects, overconfidence, and herd behavior were discovered in simple experimental designs. The field has repeatedly shown that surprisingly “simple” settings are often the most effective at revealing behavioral failures. We will emphasize this motivation more explicitly in the revision.
>
> 4. Human Incentives vs. LLM Incentives
>
> This is an important distinction. Human participants face monetary incentives and affective pressures (risk, uncertainty, FOMO, time stress) that LLMs do not experience. However, in experimental finance, even modest monetary rewards reliably produce bubble formation; professional traders also exhibit bubbles (e.g., Weitzel et al., 2019).
>
> The point of our design is not to equalize incentives, but to compare behavior under the same information, trading rules, and market structure.
>
> 5. Imperative / Bubble-Inducing Prompting
>
> We explored several prompt perturbations during development, including persona prompts, emotional framings (FOMO, trend-chasing), heuristic prompts, and model-generated variations. These altered the style of reasoning but did not produce human-scale bubbles unless the prompt explicitly said “create a bubble.”
>
> These exploratory findings suggest that the main result is not highly prompt-sensitive.
>
> 6. Questions
>
> Would larger models or RLHF variants bubble?
> This is an excellent open question. Our baseline makes it possible to detect and interpret such future deviations. We view this as an exciting direction for follow-up work.
>
> What about more complex market structures?
> The present paper focuses on the simplest diagnostic setting. Multi-asset markets, HFT environments, or asymmetric-information settings are natural extensions once the baseline is established.
>
> Does human financial knowledge matter?
> Our subject pool (university + Prolific) is standard in experimental finance. We have data on experience and demographics, and can add this in the revision.
>
> Can LLMs simulate inexperienced traders?
> Our results suggest that “out-of-the-box” LLMs behave too consistently and too close to fundamentals to mimic inexperienced humans; this is one of the conceptual takeaways of the paper.
>
> Summary
>
> We appreciate the reviewer’s positive assessment of the work. We will:
>
> Clarify why model breadth is a strength rather than a limitation,
>
> Expand discussion on why LLMs anchor to fundamentals,
>
> Emphasize the importance of the stylized design as the canonical baseline in experimental finance,
>
> More clearly delineate incentive differences.
>
> Thank you again for the constructive feedback.

---

### Official Review · Reviewer_x6wg · 2025-10-31

**Soundness:** 4
**Presentation:** 4
**Contribution:** 4
**Rating:** 6
**Confidence:** 4

**Summary:**

This paper compares the trading behavior of humans and an array of LLMs in an experimental asset market. Subjects trade the asset with each other, and the asset price depends on trading activity, allowing for the possibility of bubble formation. The paper finds that the LLMs it tests are less prone to bubble formation than human traders, this result holds both in a "mono-agent" (only copies of agents based on the same LLM) and in a "battle royale" (different LLMs competing) setting. Moreover, the two most capable LLMs tested (GPT-4o, Claude 3.5 Sonnet) exhibit the least tendency towards bubble formation. The paper also conducts a textual analysis to uncover reasons behind differences in LLM and human trading behavior.

**Strengths:**

The experimental design allows for a sensible comparison of human versus LLM behavior. Even though the LLMs tested are relatively outdated at this point, the fact that a broad array of different LLMs from different providers are tested, and the separation between human and LLM behavior is so clear, means the results strike me as credible and generalizable.

**Weaknesses:**

The textual trading strategy analysis is interesting but perhaps a little rudimentary, focusing mostly on keyword matching. Moreover, the results at the start of Section 6.1 (that the LLMs and humans all write in different styles) is not really surprising. This analysis would be strengthened by, e.g. (1) a more fine-grained semantic text analysis, (2) additional experiments in the style of Section 7, e.g. checking how LLMs' trading behavior changes if the content of the insight/plan part of the prompt changes.

**Questions:**

In Section 8, it seems like the prediction task the LLMs face is easier, because the prices in the LLM markets exhibit less volatility than the prices in the human markets. Is there possibly a way to control for this? For example, perhaps one could task each LLM with the prediction tasks of all the other markets (including the human markets).

---

### Author Response · Authors · 2025-11-20

Thank you for the thoughtful and constructive review. We appreciate the positive assessment of the paper’s soundness, presentation, and contribution, and we address the main points below.

1. Textual-Strategy Analysis

We agree that the current textual analysis is relatively coarse and can be strengthened. Our intent was to establish a first-pass comparison between human and LLM trading rationales, rather than a definitive semantic decomposition. The reviewer’s suggestions point to natural next steps.

To ensure the revision aligns with the reviewer’s expectations, we would greatly appreciate your guidance on the type of deeper analysis that would be most suitable in this context. For instance, would any of the following approaches address your concern more directly?

Embedding-based semantic clustering of trader rationales,

Predictive models (e.g., classifiers or regressors) using reasoning text to predict trade direction or deviation from FV,


Cross-model alignment metrics comparing reasoning similarity among humans and LLMs.

Because there is no established benchmark for human-vs-LLM reasoning in experimental finance, we would like your perspective on which of these (or other) approaches would strengthen the paper. We are happy to integrate a deeper semantic analysis in the revision.

2. Experiments Suggested by Section 7 (Prompt Interventions)

We agree that expanding the Section 7-style interventions would be valuable. We conducted several exploratory prompt perturbations, persona prompts, momentum/heuristic prompts, emotional framings (FOMO, “ride the trend”), and even model-generated prompts to test whether these shifts altered trading behavior. These variations did not materially change bubble formation or FV anchoring. We look forward to conducting a more systematic review of this in future work, but the initial results suggest that this behavior is robust to modest prompt changes.

3. Forecasting Task Difficulty and Volatility Differences

You correctly note that LLM-generated markets exhibit lower price volatility, which mechanically makes the forecasting task easier. We agree that this is an important point and will revise the discussion to ensure this is appropriately scoped.

Our goal in this section is not to claim that LLMs are inherently superior forecasters across all environments, but to examine how well they can anticipate both their own and others’ actions in environments where they collectively determine the price path. This is a distinctive setting: because the agents create the price dynamics, their forecasting accuracy is informative about the internal coherence of the market they generate.

In that sense, the forecasting results are interesting precisely because they reflect how LLMs behave when they dictate the underlying dynamics, a scenario that is increasingly relevant in multi-agent LLM research.

As the reviewer points out, lower volatility makes it difficult to compare absolute forecast errors across human and LLM markets directly. One could partially adjust for this using measures such as forecast errors scaled by the price standard deviation, but this would be imperfect too, because volatility differences arise endogenously from agents’ behavior—precisely the phenomenon we aim to characterize.

We will clarify in the revision that the forecasting section should be interpreted within this controlled, endogenous environment rather than as an absolute comparison of predictive skill across markets. We appreciate the reviewer raising this, and agree it is an important point for framing the contribution.

4. Outdated Models and Breadth of Tested Systems

We agree that model recency matters. In our results, the newest models (GPT-4o, Claude-Sonnet, Gemini-1.5-Pro) show the strongest adherence to fundamentals, while older models display more variability and occasional small bubbles. In other words, including older models does not generate the main result; it moves behavior slightly closer to the human pattern and therefore attenuates the effect.

We view this cross-model pattern as informative rather than a limitation. One reason this paper is useful is that it establishes a clear baseline for how current LLMs behave when prices are fully endogenous. If future, more capable models begin to diverge from FV, exhibit more human-like speculation, or shift toward momentum-driven behavior, that would itself be an interesting scientific development. Having a stylized, reproducible benchmark now is what makes such comparisons possible.

We will clarify this framing in the revision.

Summary

We appreciate the reviewer’s positive assessment and detailed suggestions. We will:

Expand the semantic depth of the text-analysis section, with guidance from the reviewer on preferred methods,

Clarify the role of model recency, and

Strengthen the explanation of how and why different textual-strategy patterns matter.

Thank you again for the thoughtful feedback.

---

> ### Comment · Reviewer_x6wg · 2025-11-26
> **Response from x6wg**
>
> Re 2+3: sounds good.
>
> Re 4: I already was on board with it being OK that the models are outdated in my original review, so sounds good.
>
> Re 1: Ultimately I am not picky about the choice of methodology, my comment here is that it would be more interesting if the text analysis focused on differentiating the strategies based on semantics. I also want to clarify my comment on additional Section-7-style experiments. I didn't mean robustness checks, I meant as a way to validate that certain textual trader rationales actually result in the trading behavior that you'd expect. The paper on LLM pricing you already build on, Fish et al. (2024), does something similar.

---

### Meta-Review · Area_Chair_kMCK · 2026-01-08

**Summary:**

The paper presents a rigorous experimental comparison of LLM and human trading behavior in a controlled market setting. Reviewers commend the novel research design and empirical rigor, highlighting significant findings that LLMs fail to reproduce key human behavioral patterns like bubble formation despite identical information and incentives. Primary concerns center on methodological scope, including the simplified single-asset market design and the potential prompt sensitivity of the results. Additionally, reviewers question whether findings from older model generations remain relevant and request clearer articulation of the paper's contribution beyond documenting LLM rationality.

**Reviewer Concerns:**

Addressed: Authors substantively engaged with critique regarding model recency and experimental design philosophy, clarifying that their baseline approach mirrors established behavioral finance methodology. They acknowledged concerns about textual analysis depth and committed to expanding semantic analysis. Outstanding: Concerns remain about whether findings translate beyond the stylized market setting, the generalizability of results to more complex real-world scenarios, and the potential impact of computational constraints on LLM behavior. One reviewer noted unresolved questions about prompt sensitivity mechanisms despite authors' informal exploration of prompt variants.

**Reviewer Scores:**

Reviewer x6wg (6): Would likely remain at marginally above threshold. The rebuttal adequately addresses textual analysis concerns and clarifies the design philosophy, but core concerns about simplified market design remain partially outstanding.
Reviewer Jkb4 (8): Likely stable at accept/poster. Authors' response satisfactorily addresses model recency questions and the value of cross-model analysis as a baseline.
Reviewer tK82 (2): Would likely remain at reject. Expressed skepticism about the fundamental contribution despite rebuttal explanations. Reviewer KABm (2): Would likely remain at reject. Concerns about experimental simplification and insufficient human participant details persist.

---

### Decision · Program_Chairs · 2026-01-26

Reject